# NeFL: Nested Federated Learning for Heterogeneous Clients

## Abstract

Federated learning (FL) is a promising approach in distributed learning keeping privacy. However, during the training pipeline of FL, slow or incapable clients (i.e., stragglers) slow down the total training time and degrade performance. System heterogeneity, including heterogeneous computing and network bandwidth, has been addressed to mitigate the impact of stragglers. Previous studies tackle the system heterogeneity by splitting a model into submodels, but with less degree-of-freedom in terms of model architecture. We propose *nested federated learning (NeFL)*, a generalized framework that efficiently divides a model into submodels using both depthwise and widthwise scaling. NeFL is implemented by interpreting forward propagation of models as solving ordinary differential equations (ODEs) with adaptive step sizes. To address the *inconsistency* that arises when training multiple submodels of different architecture, we decouple a few parameters from parameters being trained for each submodel. NeFL enables resource-constrained clients to effectively join the FL pipeline and the model to be trained with a larger amount of data. Through a series of experiments, we demonstrate that NeFL leads to significant performance gains, especially for the worst-case submodel. Furthermore, we demonstrate NeFL aligns with recent studies in FL, regarding pre-trained models of FL and the statistical heterogeneity. Code will be available after blind reviewing.

## 1 Introduction

The success of deep learning owes much to vast amounts of training data where a large amount of data comes from mobile devices and internet-of-things (IoT) devices. However, privacy regulations on data collection has become a critical concern, potentially impeding further advancement of deep learning (Dat, 2022; Dou et al., 2021). A distributed machine learning framework, federated learning (FL) is getting attention to address these privacy concerns. FL enables model training by collaboratively leveraging the vast amount of data on clients while preserving data privacy. Rather than centralizing raw data, FL collects trained model weights from clients, that are subsequently aggregated on a server by a method (e.g., FedAvg) (McMahan et al., 2017). FL has shown its potential, and several studies have explored to utilize it more practically (Hong et al., 2022; He et al., 2020b; Makhija et al., 2022; Zhuang et al., 2022; He et al., 2021).

Despite its promising perspective, there exist challenges in regarding systems-related heterogeneity (Kairouz et al., 2021; Li et al., 2020a). For example, clients with heterogeneous resources, including *computing power, communication bandwidth, and memory*, introduce stragglers that degrade the FL performance. The FL server can wait for stragglers leading to longer training times delays. Alternatively, the server can drop out a few stragglers. However, excluding stragglers can lead to a trained model biased predominantly toward data from resource-rich clients. Therefore, FL framework that accommodates clients with heterogeneous resources is required. On the other hand, one another option is to reduce a model size to accommodate resource-poor clients. However, a smaller-sized model could result in a performance degradation due to limited model capacity. To this end, FL with a single global model may not be efficient for heterogeneous clients.

In this paper, we propose *Nested Federated Learning (NeFL)*, a method that embraces existing studies of federated learning in a nested manner (Horváth et al., 2021; Diao et al., 2021; Kim et al., 2023). While generic FL trains a model with a fixed size and structure, NeFL trains several submodels of

adaptive sizes to meet dynamic requirements (e.g., memory, computing and bandwidth dynamics) of each client. We propose to scale down a model into submodels generally (by widthwise or/and depthwise). The proposed scaling method provides more degree-of-freedom (DoF) to scale down a model than previous studies (Horváth et al., 2021; Diao et al., 2021; Kim et al., 2023). The increased DoF makes submodels be more efficient in size (Tan & Le, 2019) and provides more flexibility on model size and computing cost. The scaling is motivated by interpreting a model forwarding as solving ordinary differential equations (ODEs). The ODE interpretation also motivated us to suggest the submodels with *learnable step size parameters* and the concept of *inconsistency*. We also propose a parameter averaging method for NeFL: *Nested Federated Averaging (NeFedAvg)* for averaging *consistent parameters* and FedAvg for averaging *inconsistent parameters* of submodels.

Additionally, we verify if NeFL aligns with the recently proposed ideas in FL: (i) pre-trained models improve the performance of FL in both identically independently distributed (IID) and non-IID settings (Chen et al., 2023) and (ii) simply rethinking the model architecture improves the performance, especially in non-IID settings (Qu et al., 2022). Through a series of experiments we observe that NeFL outperforms baselines sharing the advantages of recent studies.

The main contributions of this study can be summarized as follows:

- We propose a general model scaling method employing the concept of ODE solver to deal with the system heterogeneity. We introduced inconsistent parameters to deal with model-architectural discrepancies.
- We propose a method for parameter averaging across generally scaled submodels.
- We evaluate the performance of NeFL through a series of experiments and verify the applicability of NeFL over recent studies.

## 2 RELATED WORKS

**Knowledge distillation.** Knowledge distillation (KD) aims to compress models by transferring knowledge from a large teacher model to a smaller student model (Hinton et al., 2015). Several studies have explored the integration of knowledge distillation within the context of federated learning (Seo et al., 2020; Zhu et al., 2021). These studies investigate the use of KD to address (i) reducing the model size and transmitting model weights and (ii) fusing the knowledge of several models with different architectures. FedKD (Wu et al., 2022) proposes an adaptive mutual distillation where the level of distillation is controlled based on prediction performance. FedGKT (He et al., 2020a) presents an edge-computing method that employs knowledge distillation to train resource-constrained devices. The method partitions the model, and clients transfer intermediate features to an offloading server for task offloading. FedDF (Lin et al., 2020) introduces a model fusion technique that employs ensemble distillation to combine models with different architecture. However, it is worth noting that knowledge distillation-based FL requires shared data or shared generative models across clients to get the knowledge distillation loss. Alternatively, clients can transfer trained models to the other clients (Afonin & Karimireddy, 2022).

**Compression and sparsification.** Several studies have focused on compressing uploaded gradients by quantization or sparsification to deal with the communication bottleneck (Rothchild et al., 2020; Haddadpour et al., 2021). FetchSGD (Rothchild et al., 2020) proposes a method that compresses model updates using a Count Sketch. Some approaches aim to represent the weights of a larger network using a smaller network (Ha et al., 2017; Buciluă et al., 2006). Pruning is another technique that can be used to compress models. A model is pruned and retrained after the model is trained which needs additional communication and computational load for FL (Jiang et al., 2022; Mugunthan et al., 2022). A global model keeps its model size during training. Otherwise, the pruned model can be determined at the initialization (Lee et al., 2020) leading a unique pruned model with the limited capacity to be trained.

**Dynamic runtime.** The neural network can forward with larger numerical error and less computational load or forward with smaller numerical error and more computational load (Chen et al., 2018; Hairer et al., 2000). This approach offers a way to dynamically optimize computation. Split learning (Vepakomma et al., 2018; Thapa et al., 2022; He et al., 2020a) is another technique that addresses

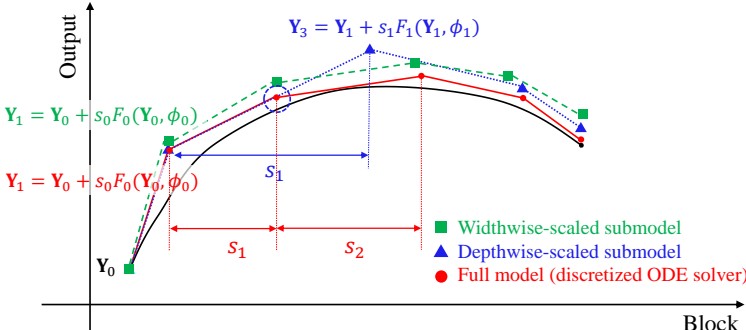

Figure 1: **Graph visualization of widthwise/depthwise model scaling inspired by ODE solver.**

the issue of resource-constrained clients by leveraging richer computing resources, such as cloud or edge servers. These methods enable clients with system-related heterogeneity to participate in the FL pipeline.

**Model splitting.** Various approaches have been proposed to address the heterogeneity of clients by splitting global network based on their capabilities. LG-FedAvg (Liang et al., 2020) introduced a method to split the model and decouple layers into global and local layers, reducing the number of parameters involved in communication. While FjORD (Horváth et al., 2021) and HeteroFL (Diao et al., 2021) split a global model widthwise, DepthFL (Kim et al., 2023) splits a global model depthwise. Unlike widthwise scaling, DepthFL incorporates an additional bottleneck layer and an independent classifier for every submodel. It has been studied that *deep and narrow* models as well as *shallow and wide* models are inefficient in terms of the number of parameters or floating-point operations (FLOPs). Prior studies have shown that carefully balancing the depth and width of a network can lead to improved performance (Zagoruyko & Komodakis, 2016; Tan & Le, 2019). Therefore, a balanced network should be considered for FL that splits a global model into submodels. Our proposed NeFL provides a scaling method both widthwise and depthwise for submodels to be well-balanced.

## 3 BACKGROUND

We propose to scale down the models inspired by solving ODEs in a numerical way (e.g., Euler method). Modern deep neural networks stack residual blocks that contain skip-connections that bypass the residual layers. A residual block is written as $\mathbf{Y}_{j+1} = \mathbf{Y}_j + F_j(\mathbf{Y}_j, \phi_j)$, where $\mathbf{Y}_j$ is the feature map at the $j$th layer, $\phi_j$ denotes the $j$th block's network parameters, and $F_j$ represents a residual module of the $j$th block. These networks can be interpreted as solving ODEs by numerical analysis (Chang et al., 2018; He et al., 2016).

Consider an initial value problem to find $y$ at $t$ given $\frac{dy}{dt} = f(t, y)$ and $y(t_0) = y_0$. We can obtain $y$ at any point by integration: $y = y_0 + \int_{t_0}^{t} f(t, y)dt$. It can be approximated by Taylor's expansion as $y = y_0 + f(t_0, y_0)(t - t_0)$ and approximated after more steps as follows:

$$y_{n+1} = y_n + hf(t_n, y_n) = y_0 + hf(t_0, y_0) + \cdots + hf(t_{n-1}, y_{n-1}) + hf(t_n, y_n), \quad (1)$$

where $h$ denotes the step size. An ODE solver can compute with less steps by using larger step size as: $y_{n+1} = y_0 + 2hf(t_0, y_0) + \cdots + 2hf(t_{n-1}, y_{n-1})$ when $n$ is odd. The results would be numerically less accurate than fully computing with a smaller step size. Note that the equation looks like the equation of residual connections. An output of a residual block is rewritten as follows:

$$\mathbf{Y}_{j+1} = \mathbf{Y}_j + F_j(\mathbf{Y}_j, \phi_j) = \mathbf{Y}_0 + F_0(\mathbf{Y}_0, \phi_0) + \cdots + F_{j-1}(\mathbf{Y}_{j-1}, \phi_{j-1}) + F_j(\mathbf{Y}_j, \phi_j). \quad (2)$$

Motivated by this interpreting the neural networks as solving ODEs, we proposed that few residual blocks can be omitted during forward propagation. For example, $\mathbf{Y}_3 = \mathbf{Y}_0 + F_0 + F_1 + F_2$ could be approximated by $\mathbf{Y}_3 = \mathbf{Y}_0 + F_0 + 2F_1$ omitting the $F_2$ block. Here, we propose *learnable step size parameters* (Touvron et al., 2021; Bachlechner et al., 2021). Instead of pre-determining the step size parameters ($h$'s in the equation (1)), we let step size parameters be trained along with the network parameters. For example, when we omit $F_2$ block, output is formulated as $\mathbf{Y}_3 = \mathbf{Y}_0 + s_0 F_0 + s_1 F_1$

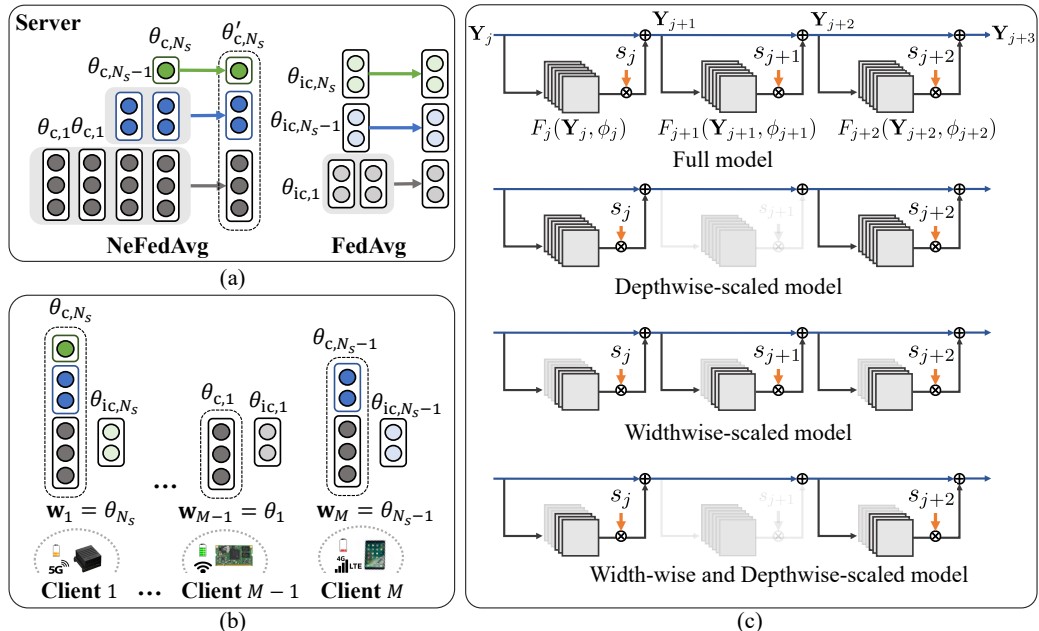

Figure 2: **FL with generally split submodels**. NeFL aggregates weights of submodels, where the submodels are scaled by both widthwise and depthwise. The weights include consistent and inconsistent parameters.

where $s_i$'s are also optimized. This enables to scale down the network depthwise. Furthermore, it is also possible to scale each residual block by width (e.g., the size of filters in convolutional layers). The theoretical background behind width-wise scaling is provided by the Eckart-Young-Mirsky theorem (Eckart & Young, 1936). A width-wise scaled residual block represents the optimal $k$-rank approximation of the original (full) block (Horváth et al., 2021).

Our model scaling method inpired by ODE solver is displayed in Figure 1. The black line represents a function to approximate, while the red line represents output approximated by a full nerual netowk. The blue color line represents the depthwise-scaled submodel. The model omitted to compute block $F_2(\cdot)$ at the point $\mathbf{Y}_2$ and instead, larger step size for computing $F_1(\mathbf{Y}_1, \phi_1)$ compensates the omitted block (Chang et al., 2018). The green colored line represents the widthwise-scaled submodel that has less parameters $\phi$ in each block. Following the theorem that a widthwise-scaled model with less parameters is an approximation of a model with more parameters (Eckart & Young, 1936), output from a widthwise-scaled model has larger numerical error.

## 4 NeFL: Nested Federated Learning

*NeFL* is FL framework that accommodates resource-poor clients. Instead of requiring every client to train a single global model, NeFL allows resource-poor clients to train submodels that are scaled in both widthwise and depthwise (Figure 2c), based on dynamic nature of their environments. This flexibility enables more clients, even with constrained resources, to pariticipate in the FL pipeline, thereby making the model be trained with a more data. Referring to Algorithm 1, in NeFL pipeline, NeFL server broadcast the weights to clients that heterogeneous clients have an ability to determine their submodel at every iteration (Figure 2b), allowing them to adapt to varying computing or communication bottlenecks in their respective environments. The NeFL server subsequently aggregates the parameters of these different submodels, resulting in a collaborative and comprehensive global model. Furthermore, during test-time[1], clients have a flexibility to select an appropriate submodel based on their specific background process burdens, such as runtime memory constraints or CPU/GPU bandwidth dynamics. This allows clients to balance between performance and factors like memory usage or latency, tailoring to their individual needs and constraints. To this end, NeFL is twofold: scaling a model into submodels and aggregating the parameters of submodels.

---

[1]At test-time, trained submodels are utilized for inference.

---

**Algorithm 1** NeFL: Nested Federated Learning

---

**Input:** Submodels $\{\mathbf{F}_k\}_{k=1}^{N_s}$, total communication round $T$, the number of local epochs $E$
 1: **for** $t \leftarrow 0$ to $T - 1$ **do** // Global iterations
 2:     NeFL server broadcasts the weights $\{\theta_{c,N_s}, \theta_{ic,1}, \ldots \theta_{ic,N_s}\}$ to clients in $\mathcal{C}_t$
 3:     $\mathbf{w}_i \leftarrow \theta_j \in \{\theta_1, \ldots, \theta_{N_s}\} \, \forall i \in \mathcal{C}_t$
 4:     **for** $k \leftarrow 0$ to $E - 1$ **do** // Local iterations
 5:         Client $i$ updates $\mathbf{w}_i \, \forall i \in \mathcal{C}_t$
 6:     Client $i$ transmits the weights $\mathbf{w}_i$ to the NeFL server $\forall i \in \mathcal{C}_t$
 7:     $\{\theta_{c,N_s}, \theta_{ic,1}, \ldots \theta_{ic,N_s}\} \leftarrow \texttt{ParameterAverage}(\{\mathbf{w}_i\}_{i \in \mathcal{C}_t})$ ▷ Algorithm 2

---

NeFL scales a global model $\mathbf{F}_G$ into $N_s$ submodels $\mathbf{F}_1, \ldots, \mathbf{F}_{N_s}$ with corresponding weights $\theta_1, \ldots, \theta_{N_s}$. Without loss of generality, we suppose $\mathbf{F}_G = \mathbf{F}_{N_s}$. The scaling method is described in following Section 4.1. During each communication round $t$, each client in subset $\mathcal{C}_t$ (which is subset of $M$ clients) selects one of the $N_s$ submodels based on their respective environments and trains the model by the local epochs $E$. Then, clients transmit their trained weights $\{\mathbf{w}_i\}_{i \in \mathcal{C}_t}$ to the NeFL server, which aggregates them into $\{\theta_{c,N_s}, \theta_{ic,1}, \ldots \theta_{ic,N_s}\}$ where $\theta_{c,k}$ and $\theta_{ic,k}$ denote *consistent* and *inconsistent* parameters of a submodel $k$. Note that $\theta_{c,k} \subset \theta_{c,N_s} \, \forall k$. The aggregated weights are then distributed to the clients in $\mathcal{C}_{t+1}$, and this process continues for the total number of communication rounds.

## 4.1 MODEL SCALING

We propose a global network scaling method, which combines both widthwise scaling and depthwise scaling. We scale the model widthwise by ratio of $\gamma_W$ and depthwise by ratio of $\gamma_D$. For example, a global model is represented as $\gamma = \gamma_W \gamma_D = 1$ and a submodel that has 25% parameters of a global model can be scaled by $\gamma_W = 0.5$ and $\gamma_D = 0.5$. Flexible widthwise/depthwise scaling provides more degree of freedom to split models. We further describe on the scaling strategies in the following sections.

### 4.1.1 DEPTHWISE SCALING

Residual networks, such as ResNets (He et al., 2016) and ViTs (Dosovitskiy et al., 2021), have gained popularity due to their significant performance improvements on *deep* networks. The output of a residual block can be seen as a function multiplied by a step size as $\mathbf{Y}_{j+1} = \mathbf{Y}_j + s_j F(\mathbf{Y}_j, \phi_j) = \mathbf{Y}_0 + \sum_j s_j F(\mathbf{Y}_j, \phi_j)$. Note that ViTs' forward operation with skip connections can be represented in a similar way by choosing $F(\cdot)$ as either self-attention (SA) layers or feed-forward networks (FFN):

$$\mathbf{Y}_{j+1} = \mathbf{Y}_j + s_j SA(\mathbf{Y}_j, \phi_j), \quad \mathbf{Y}_{j+2} = \mathbf{Y}_{j+1} + s_{j+1} FFN(\mathbf{Y}_{j+1}, \phi_{j+1}).$$

The depthwise scaling can be implemented by skipping any of residual blocks of ResNets or encoder blocks of ViTs. For example, ResNets (e.g., ResNet18, ResNet34) have a downsampling layer followed by residual blocks, each consisting of two convolutional layers. ResNet18 has 8 residual blocks and ResNet34 has 16 residual blocks. ViT/B-16 has 12 encoder blocks where embedding patches are inserted as inputs to following transformer encoder blocks. Each block can have different number of the parameters and computational complexity. A submodel is scaled by omitting few blocks satisfying the number of parameters to be scaled by $\gamma_D$ according to system requirements of a client. We can observe that skipping a few blocks in a network still allows it to operate effectively (Chang et al., 2018). This observation is in line with the concept of stochastic depth proposed in (Huang et al., 2016), where a subset of blocks is randomly bypassed during training and the all of the blocks are used during testing.

The step size parameters $s$'s can be viewed as dynamically training how much forward pass should be made at each step. They allow each block to adaptively contribute to useful representations, and larger step sizes are multiplied to blocks that provide valuable information. Referring to Figure 1 and numerical analysis methods (e.g., Euler method, Runge-Kutta methods (Hairer et al., 2000)), adaptive step sizes rather than uniformly spaced steps, can effectively reduce numerical errors. *Note that DepthFL (Kim et al., 2023) is a special case of the proposed depthwise scaling.*

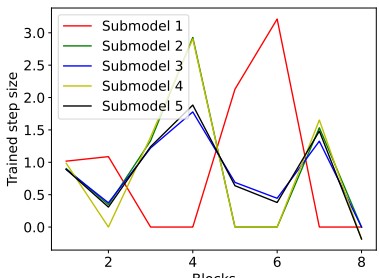 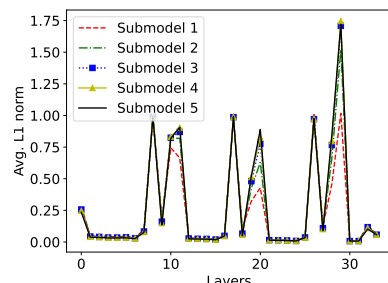

(a) The trained submodels have different step sizes.    (b) Average L1 norm of weights of submodels trained.

Figure 3: Weights of trained five submodels by NeFL

### 4.1.2 WIDTHWISE SCALING

Previous studies have been conducted on reducing the model size in the width dimension (Li et al., 2017; Liu et al., 2017). In the context of NeFL, widthwise scaling is employed to slim down the network by utilizing structured contiguous pruning (ordered dropout in (Horváth et al., 2021)) along with learnable step sizes. We apply contiguous channel-based pruning to the convolutional networks (He et al., 2017) and node removal to the fully connected layers. The parameters of narrow-width submodel constitute a subset of the parameters of the wide-width model. Consequently, parameters of the slimmest model are trained on any submodel by every client, while the parameters of the larger model is trained less frequently. This approach ensures that the parameters of the slimmest model capture the most of useful representations. Note that each block stated in Section 4.1.1 can have different width, but we suppose throughout the paper that every block has same widthwise scaling ratio $\gamma_W$ without loss of generality.

We can obtain further insights from a toy example. Given an optimal linear neural network $y = Ax$ and data from uniform distribution $x \sim \mathcal{X}$, the optimal widthwise-scaled submodel $y = A_W x$ is the best $k$-rank approximation[2] by Eckart–Young–Mirsky theorem (Horváth et al., 2021; Eckart & Young, 1936). Given a linear neural network of of rank $k$, widthwise-scaled model of scaling ratio $\gamma_W$ has rank of $\lceil \gamma_W k \rceil$. Then, $\min \mathbb{E}_{x \sim \mathcal{X}} \|A_W x - Ax\|_F^2 = \min \|A_W - A\|_F^2$, where $F$ denotes Frobenius norm. In our framework, similar tedency is observed. Inspired by the magnitude-based pruning (Han et al., 2015; Li et al., 2017), we present the L1 norm of weights averaged by the number of weights at each layer of five trained submodels in Figure 3b. The submodel 1 which is the slimmest model, has similar tendency of L1 norm to the widest model and the gap between submodel gets smaller as the model size gets wider. The slimmest model might have learned the most useful representation while additional parameters for larger models obtain still useful, but less useful information.

### 4.2 PARAMETER AVERAGING

**Inconsitency.** NeFL enables training several submodels on local data for subsequent aggregation. However, submodels of different model architectures (i.e., different width and depth) have different characteristics (Acar et al., 2021; Chen et al., 2022; Li et al., 2020b; Santurkar et al., 2018). Referring to Figure 1, if depthwise-scaled model omits a block, it can be compensated by optimizing step size of adjacent blocks. For widthwise-scaled model, numerical errors are induced from each block and they can be mitigated by optimizing step size for each block. We can infer that each submodel requires different step sizes to compensate for the numerical errors according to its respective model architecture. Furthermore, submodels with different model architectures have different loss landscapes. Consider the situation where losses are differentiable. A stationary point of a global model is not necessarily the stationary points of individual submodels (Acar et al., 2021; Li et al., 2020b). Different trainability of neural networks, which varies depending on network architecture, may lead the convergence of submodels to become non-stationary.

---

[2] Note that we have learnable step sizes that can further improve the performance of widthwise scaled submodel over best $k$-rank approximation (refer to Appendix C).

---

**Algorithm 2** `ParameterAverage`

---

**Input:** Trained weights from clients $\mathbf{W} = \{\mathbf{w}_i\}_{i \in \mathcal{C}_t}$

1: **for** submodel index $k$ in $\{1, \ldots, N_s\}$ **do**
2:     $\mathcal{M}_k \leftarrow \{\mathbf{w}_i | \mathbf{w}_i = \theta_k\}_{i \in \mathcal{C}_t}$
3: **for** block $\phi_j$ in $\theta_{c,N_s}$ **do** // `NeFedAvg`                ▷ Consistent parameters
4:     $\mathcal{M}' \leftarrow \mathcal{M} = \{\mathcal{M}_k\}_{k=1}^{N_s}$
5:     **for** $k$ in $\{1, \ldots, N_s\}$ **do**
6:         $\mathcal{M}' \leftarrow \mathcal{M}' \setminus \mathcal{M}_k$ if $\phi_j \notin \theta_{c,k}$
7:     $k' \leftarrow 0, \phi_{j,0} = \emptyset$
8:     **for** $k$ in $\{k | \mathcal{M}_k \in \mathcal{M}'\}$ **do**
9:         $\phi_{j,k} \setminus \phi_{j,k'} \leftarrow \sum_{\{i | \mathbf{w}_i \in \bigcup_{l \geq k, \mathcal{M}_l \in \mathcal{M}'} \mathcal{M}_l\}} \phi_{j,k}^i \setminus \phi_{j,k'}^i / \sum_{l \geq k, \mathcal{M}_l \in \mathcal{M}'} |\mathcal{M}_l|$
10:         $k' \leftarrow k$
11: $\theta_{c,N_s} \leftarrow \bigcup_j \phi_j$
12: **for** $k$ in $\{1, \ldots, N_s\}$ **do** // `FedAvg` (McMahan et al., 2017)       ▷ Inconsistent parameters
13:     $\theta_{ic,k} \leftarrow \sum_{\{i | \mathbf{w}_i \in \mathcal{M}_k\}} \theta_{ic,k}^i / |\mathcal{M}_k|$

---

The motivation led us to introduce a decoupled set of parameters (Liang et al., 2020; Arivazhagan et al., 2019) for respective submodels that require different parameter averaging method. We refer to these different characteristics between submodels as *inconsistency*. To this end, we propose the concept of separating a few parameters, referred to as *inconsistent parameters*, from individual submodels. We address step size parameters and batch normalization layers as inconsistent parameters. Note that batch normalization can improve Lipschitzness of loss, which is sensitive to the convergence of FL (Santurkar et al., 2018; Li et al., 2020c). Figure 3a presents the trained step size parameters of submodels. We further demonstrate the benefits of inconsistent parameters by ablation study in Appendix A.

Meanwhile, *consistent parameters* are averaged across submodels. Since the weights of a submodel are subset of the weights of the largest submodel, the averaging of these weights should be different from conventional FL. The parameters of the submodels are denoted as $\theta_1 = \{\theta_{c,1}, \theta_{ic,1}\}, \ldots, \theta_{N_s} = \{\theta_{c,N_s}, \theta_{ic,N_s}\}$ where $\theta_c$ denotes consistent parameters and $\theta_{ic}$ denotes inconsistent parameters. Note that the global parameters, which are broadcasted to clients for the next FL iteration, encompass $\theta_{c,N_s}, \theta_{ic,1}, \ldots, \theta_{ic,N_s}$. Note that $\theta_{c,k} \subset \theta_{c,N_s} \forall k$.

**Parameter averaging.** We propose averaging method for the uploaded weights from clients in Algorithm 2 (Figure 2a). Locally trained weights $\mathbf{W} = \{\mathbf{w}_i\}_{i \in \mathcal{C}_t}$ from clients are provided as input. The NeFL server verifies the correspondence of each client's updated weights to specific submodels and stores these weights separately for each submodel. $\mathcal{M} = \{\mathcal{M}_1, \ldots, \mathcal{M}_{N_s}\}$, where $\mathcal{M}_k$ denotes weights set from clients who trained $k$-th submodel, is set of uploaded weights sorted by submodels. The consistent parameters are averaged in a nested manner (*Nested Federated Averaging (NeFedAvg)*). The parameters are averaged by weights from clients who trained the parameters in this round $t$. For NeFedAvg, the server accesses parameters block by block ($\phi_j$). For each block, the server checks which submodel has the block (line 6 in Algorithm 2). Then, parameters are averaged by width in a nested manner (line 9 in Algorithm 2). The parameters of a block with the smallest width are included in the parameters of a block with larger width. Hence, the block parameters are averaged by weights of clients ($\phi_{j,k}^i$; $j$-th block parameters of $k$-th submodel that $i$-th client updated) who updated the parameters. For averaging inconsistent parameters, FedAvg (McMahan et al., 2017) is employed. Each submodel has the same size of inconsistent parameters that the inconsistent parameters are averaged for respective submodels. Note that $\theta_{ic,k}^i$ denotes the inconsistent parameters of $k$-th submodel that $i$-th client has updated. The example for Algorithm 2 is provided in Appendix B.3.

## 5 EXPERIMENTS

In this section, we demonstrate the performance of our proposed NeFL over baselines. We provide the experimental results of NeFL that trains submodels from scratch. We also demonstrate whether NeFL aligns with the recently proposed ideas in FL through experiments to verify (i) the perfor-

Table 1: Results of NeFL for CIFAR-10 dataset under **IID** (left) and **non-IID** (right) settings are presented: Top-1 classification accuracies (%) for the worst-case submodel and the average of the performance of five submodels.

| Model | Method | IID | | non-IID | |
|---|---|---|---|---|---|
| | | Worst | Avg | Worst | Avg |
| ResNet18 | HeteroFL | 80.62 ($\pm$ 0.24) | 84.26 ($\pm$ 1.95) | 76.25 ($\pm$ 1.05) | 80.11 ($\pm$ 2.03) |
| | FjORD | 85.12 ($\pm$ 0.22) | 87.32 ($\pm$ 1.21) | 75.81 ($\pm$ 5.65) | 77.99 ($\pm$ 6.50) |
| | DepthFL | 64.80 ($\pm$ 10.49) | 82.44 ($\pm$ 10.17) | 59.61 ($\pm$ 5.16) | 76.89 ($\pm$ 9.60) |
| | **NeFL (ours)** | **86.86 ($\pm$ 0.22)** | **87.88 ($\pm$ 0.68)** | **81.26 ($\pm$ 2.44)** | **81.71 ($\pm$ 3.14)** |
| ResNet34 | HeteroFL | 79.51 ($\pm$ 0.44) | 83.16 ($\pm$ 1.96) | 76.03 ($\pm$ 1.34) | 79.63 ($\pm$ 5.24) |
| | FjORD | 85.12 ($\pm$ 0.25) | 87.36 ($\pm$ 1.19) | 74.70 ($\pm$ 3.66) | 76.01 ($\pm$ 5.24) |
| | DepthFL | 25.73 ($\pm$ 4.25) | 75.30 ($\pm$ 24.88) | 30.42 ($\pm$ 9.34) | 70.76 ($\pm$ 21.04) |
| | **NeFL (ours)** | **87.71 ($\pm$ 0.37)** | **89.02 ($\pm$ 0.80)** | **80.76 ($\pm$ 2.82)** | **83.31 ($\pm$ 2.94)** |

Table 2: Results of NeFL employing **pre-trained** models as initial weights for CIFAR-10 dataset under **IID** (left) and **non-IID** (right) settings are presented: Top-1 classification accuracies (%) for the worst-case submodel and the average of the performance of five submodels.

| Model | Method | IID | | non-IID | |
|---|---|---|---|---|---|
| | | Worst | Avg | Worst | Avg |
| Pre-trained ResNet18 | HeteroFL | 78.26 ($\pm$ 0.15) | 84.48 ($\pm$ 3.04) | 71.95 ($\pm$ 1.32) | 76.17 ($\pm$ 3.39) |
| | FjORD | 86.37 ($\pm$ 0.18) | 88.91 ($\pm$ 1.37) | 81.81 ($\pm$ 1.10) | 81.96 ($\pm$ 5.76) |
| | DepthFL | 47.76 ($\pm$ 8.54) | 82.86 ($\pm$ 17.98) | 39.78 ($\pm$ 3.74) | 67.71 ($\pm$ 16.88) |
| | **NeFL (ours)** | **88.61 ($\pm$ 0.08)** | **89.60 ($\pm$ 0.70)** | **82.91 ($\pm$ 0.47)** | **85.85 ($\pm$ 2.43)** |
| Pre-trained ResNet34 | HeteroFL | 79.97 ($\pm$ 0.53) | 84.34 ($\pm$ 2.33) | 72.33 ($\pm$ 1.59) | 78.2 ($\pm$ 3.29) |
| | FjORD | 87.08 ($\pm$ 0.29) | 89.37 ($\pm$ 1.30) | 78.2 ($\pm$ 4.39) | 78.90 ($\pm$ 6.23) |
| | DepthFL | 52.08 ($\pm$ 5.30) | 83.63 ($\pm$ 15.97) | 42.09 ($\pm$ 2.79) | 79.86 ($\pm$ 18.13) |
| | **NeFL (ours)** | **88.36 ($\pm$ 0.11)** | **91.14 ($\pm$ 1.42)** | **83.62 ($\pm$ 0.68)** | **86.48 ($\pm$ 2.18)** |

mance of NeFL with initial weights loaded from pre-trained model (Chen et al., 2023) and (ii) the performance of NeFL with ViTs in non-IID settings (Qu et al., 2022). We conduct experiments using the CIFAR-10 dataset (Krizhevsky et al.) for image classification with ResNets and ViTs.

**Experimental setup.** The experiments in Table 1 and Table 2 are evaluated with five submodels ($N_s = 5$ where $\gamma_1 = 0.2, \gamma_2 = 0.4, \gamma_3 = 0.6, \gamma_4 = 0.8, \gamma_5 = 1$) and the experiments in Table 3 are evaluated with three submodels ($N_s = 3$ where $\gamma_1 = 0.5, \gamma_2 = 0.75, \gamma_3 = 1$). Pre-trained models we use for evaluation are trained on the ImageNet-1k dataset(Deng et al., 2009; Pyt, 2023). The pre-trained weights trained on ImageNet-1k dataset (Deng et al., 2009) are loaded on the initial global models and subsequently NeFL was performed. To take system heterogeneity into account, each client is assigned one of the submodels at each iteration, and statistical heterogeneity was implemented by label distribution skew following the Dirichlet distribution with concentration parameter 0.5 (Yurochkin et al., 2019; Li et al., 2021a). Training details are provided in Appendix B.1.

**Comparison with state-of-the-art model splitting FL methods.** For fair comparison across different baselines, we designed each submodel to have similar number of parameters (Table 8 in Appendix A). As illustrated in Table 1, NeFL outperforms baselines in terms of both the performance of the worst-case submodel ($\gamma = 0.2$) and the average performance across five submodels in IID and non-IID settings. Notably, the performance gain is greater in non-IID settings, which belong to practical FL scenarios. It is worth noting that our proposed depthwise scaling method has performance gain over depthwise scaling baselines, while our proposed widthwise scaling method also has performance gain over widthwise scaling baselines (refer to the Appendix A). Furthermore, beyond the performance gain from using depthwise or widthwise scaled submodels, NeFL provides a federated averaging method that can incorporate widthwise or/and depthwise scaled submodels.

Table 3: Results of NeFL with three submodels for CIFAR-10 dataset under **IID** (left) and **non-IID** (right) settings. A initial weights for global model was given with the pre-trained model with ImageNet-1k. We report Top-1 classification accuracies (%) for the submodels.

| Model | Param. # | IID | | non-IID | |
|---|---|---|---|---|---|
| | | Worst | Avg | Worst | Avg |
| Pre-trained ViT | 86.4M | 93.02 ($\pm$ 0.06) | 95.96 ($\pm$ 2.10) | 87.56 ($\pm$ 0.16) | 92.74 ($\pm$ 3.95) |
| Pre-trained Wide ResNet101 | 124.8M | 90.9 ($\pm$ 0.16) | 91.35 ($\pm$ 0.39) | 87.17 ($\pm$ 0.04) | 87.74 ($\pm$ 1.06) |

This characteristic of embracing any submodel with different architecture extracted from a single global model enhances flexibility, enabling more clients to participate in the FL pipeline.

**Performance enhancement by employing pre-trained models.** We investigate the performance improvement from incorporating pre-trained models into NeFL and verify that NeFL is still effective when employing pre-trained models. Recent studies on FL have figured out that FL gets benefits from pre-trained models even more than centralized learning (Kolesnikov et al., 2020; Chen et al., 2023). It motivates us to evaluate the performance of NeFL on pre-trained models. The pre-trained model that is trained in a common way using ImageNet-1k is loaded from PyTorch (Paszke et al., 2019). Even when *a pre-trained model was trained as a full model without any submodel being trained*, NeFL made better performance with these pre-trained models. The results in Table 2 show that the performance of NeFL has been enhanced through pre-training in both IID and non-IID settings following the results of the recent studies. Meanwhile, baselines such as HeteroFL and DepthFL, which do not have any inconsistent parameters, have no effective performance gain when trained with pre-trained models compared to to models trained from scratch.

**Impact of model architecture on statistical heterogeneity.** We now present an experiment using ViTs and Wide ResNet (Zagoruyko & Komodakis, 2016) on NeFL. Previous studies have examined the effectiveness of ViTs in FL scenarios, and it has been observed that ViTs can effectively alleviate the adverse effects of statistical heterogeneity due to their inherent robustness to distribution shifts (Qu et al., 2022). Building upon this line of research, Table 3 demonstrates that ViTs outperform ResNets in our framework, with the larger number of parameters, in both IID and non-IID settings. Particularly in non-IID settings, ViTs exhibit less performance degradation of average performance when compared to IID settings. Note that when comparing the performance gap between IID and non-IID settings, the worst-case ViT submodel experiences more degradation than the worst-case ResNet submodel. Nevertheless, despite this degradation, ViT still maintains higher performance than ResNet. Consequently, we verify that ViT on NeFL is also effective following the results in Qu et al. (2022).

**Additional experiments.** We provide experimental results of NeFL on other datasets and with different number of clients, along with ablation study in Appendix A. The performance gain of NeFL increases when dealing with more challenging datasets. For example, the performance gain of NeFL with ResNet34 on CIFAR-100 is 7.63% over baselines. We also verify that NeFL shows the best performance across all different number of clients. In our ablation study, we present the effectiveness of inconsistent parameters including learnable step sizes and the performance comparison of proposed scaling methods.

## 6 CONCLUSION

In this work, we have introduced *Nested Federated Learning (NeFL)*, a generalized FL framework that addresses the challenges of system heterogeneity. By leveraging proposed depthwise and widthwise scaling, NeFL efficiently divides models into submodels employing the concept of ODE solver, leading to improved performance and enhanced compatibility with resource-constrained clients. We propose to decouple few parameters as inconsistent parameters for respective submodels that FedAvg are employed for averaging inconsistent parameters while NeFedAvg was utilized for averaging consistent parameters. Our experimental results highlight the significant performance gains achieved by NeFL, particularly for the worst-case submodel. Furthermore, we also explore NeFL in line with recent studies of FL such as pretraining and statistical heterogeneity.

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
