# Supplementary Material

## A ADDITIONAL EXPERIMENTS

### A.1 OTHER DATASET

We evaluate the performance for other dataset such as CIFAR-100 (Krizhevsky et al.), CINIC-10 (Darlow et al., 2018) SVHN (Netzer et al., 2011) and we observe a similar tendency in terms of Top-1 accuracy of the worst-case submodel and average accuracy over submodels. Note that we set total communication round $T = 100$ for training SVHN. The results are presented in Table 4.

Table 4: Results of NeFL with five submodels for **CIFAR-100** (left), **CINIC10** (center) and **SVHN** (right) dataset under IID settings. We report Top-1 classification accuracies (%) for the worst-case submodel and the average of the performance of five submodels.

| Model | Method | CIFAR-100 | | CINIC-10 | | SVHN | |
|-------|--------|-----------|------|----------|------|------|------|
| | | Worst | Avg | Worst | Avg | Worst | Avg |
| ResNet18 | HeteroFL | 41.33 | 47.09 | 67.55 | 70.40 | 91.82 | 93.46 |
| | FjORD | 49.29 | 52.67 | 71.95 | 74.98 | 94.31 | 93.97 |
| | DepthFL | 31.68 | 49.56 | 54.51 | 71.42 | 91.54 | 93.97 |
| | **NeFL (ours)** | **52.63** | **53.62** | **74.16** | **75.29** | **94.45** | **94.94** |
| ResNet34 | HeteroFL | 34.96 | 39.75 | 67.39 | 69.62 | 89.86 | 92.39 |
| | FjORD | 47.59 | 50.7 | 71.58 | 74.19 | 93.83 | 94.63 |
| | DepthFL | 14.51 | 46.79 | 32.05 | 67.04 | 74.33 | 89.96 |
| | **NeFL (ours)** | **55.22** | **56.26** | **75.02** | **76.68** | **94.72** | **95.22** |

### A.2 DIFFERENT NUMBER OF CLIENTS

We conduct further experiments across different numbers of clients. In Table 5, we observe that as the number of clients increases, the performance of NeFL as well as baselines degrades. The results align with previous studies (Kim et al., 2023; Thapa et al., 2022; Wang et al., 2020). The more the number of clients, trained weights deviates further from the weights trained by IID data. While the IID sampling of the training data ensures the stochastic gradient to be an unbiased estimate of the full gradient, the non-IID sampling leads to non-guaranteed convergence and model weight divergence in FL (Li et al., 2020b; 2021b; Zhao et al., 2018). The local clients train their own network with multiple epochs and upload the weights so that the uploaded weights get more deviated. In this regard, our proposed algorithm remains effective across different numbers of clients; however, the performance (e.g., accuracy and convergence) degrades by the data distribution among clients varies more as their number increases.

### A.3 ABLATION STUDY

In this section, we provide ablation study for evaluating the performance gains of both width/depthwise scaling, inconsistent parameters and learnable step size parameters. *NeFL-W* denotes that all submodels are scaled widthwise, *NeFL-D* denotes that all submodels are scaled depthwise and *NeFL-WD* that submodels are scaled both widthwise and depthwise. We further refer to NeFL-D$_O$ that has different initial step sizes with NeFL-D. Referring to Table 12, NeFL-D$_O$ has larger magnitude step sizes aligning with the principles of ODE solver, compared to NeFL-D. NeFL-D scales submodels by skipping a subset of blocks of a global model, thus reducing the depth of the model. NeFL-D does not compensate for the skipped blocks by using larger step sizes. For example, a submodel in NeFL-D is given the initial step sizes as $s_0 = 1, s_1 = 1, s_2 = 0$ and output after Block 2 without Block 2 is $\mathbf{Y}_3 = \mathbf{Y}_0 + F_0 + F_1$. Meanwhile, NeFL-D$_O$ reduces the size of the global model by skipping a subset of block functions $F(\cdot)$ and gives larger initial step sizes to compensate it. The step sizes are determined based on the number of blocks that are skipped. For a

Table 5: Results of NeFL of five submodels with a global model ResNet18 for CIFAR-10 dataset under IID settings across different number of clients. We report Top-1 classification accuracies (%) for the worst-case submodel and the average of the performance of five submodels..

| # of Clients | Model size | Method | | | |
|---|---|---|---|---|---|
| | | **NeFL (ours)** | FjORD | HeteroFL | DepthFL |
| 100 | Worst | **86.86** | 85.12 | 80.62 | 64.8 |
| | Avg | **87.88** | 87.32 | 84.62 | 82.44 |
| 50 | Worst | **88.42** | 86.19 | 84.67 | 52.07 |
| | Avg | **89.14** | 88.43 | 87.23 | 82.04 |
| 20 | Worst | **89.2** | 87.76 | 88.74 | 24.94 |
| | Avg | **89.88** | 89.6 | 88.71 | 76.54 |

submodel without Block 2, initial output after Block 2 is initially computed as $\mathbf{Y}_3 = \mathbf{Y}_0 + F_0 + 2F_1$ given $s_0 = 1, s_1 = 2, s_2 = 0$. We also refer that submodel with no learnable step sizes by N/L (i.e., constant step sizes are kept with given intial values).

The performance comparison between NeFL-WD and NeFL-WD (N/L) as well as the comparison between NeFL-W and FjORD (Horváth et al., 2021) provides the effectiveness of learnable step sizes and comparison between NeFL-W and HeteroFL (Diao et al., 2021) provides the effectiveness of inconsistent parameters including learnable step sizes. Similarly, the comparison between NeFL-D and NeFL-D (N/L) provides the effectiveness of learnables step sizes and comparison between NeFL-D and DepthFL (Kim et al., 2023) provides the effectiveness of inconsistent parameters including learnable step sizes. We summarized the NeFL with various scaled submodels in Table 7. We also provide the parameter sizes and average FLOPs of submodels by scaling in Table 8.

**Learnable step size parameters & inconsistent parameters.** Referring to the Table 6, the performance improvements of NeFL-WD over NeFL-WD (N/L), NeFL-W over FjORD (Horváth et al., 2021) and NeFL-D over NeFL-D (N/L) provide *the effectiveness of learnable step sizes. The effectiveness of the inconsistent parameters including learn step sizes* is also verified by NeFL-D over DepthFL (Kim et al., 2023) and NeFL-W over HeteroFL (Diao et al., 2021).

**Scaling method.** We also observe that NeFL-D and NeFL-WD have better performance over widthwise scaling. The performance gap of depthwise scaling over widthwise scaling gets larger for narrow and deep networks. Note that ResNet56 and ResNet110 has smaller (i.e., narrower) channel sizes with more layers (i.e., deeper) than ResNet18 and ResNet34 He et al. (2016). Furthermore, we have a finding that *NeFL-D outperforms NeFL-$D_O$ in most cases*. The rationale comes from the empirical results that trained step sizes are not as large as initial value for NeFL-$D_O$ that large initial values for NeFL-$D_O$ degrades the trainability of depthwise-scaled submodels.

We evaluated the experiments by similar number of parameters for several scaling methods and depthwise scaling requires slightly more FLOPs than widthwise scaling for ResNet18, ResNet34 and ResNet110 and less FLOPs for ResNet56. Usually depthwise scaled models have more FLOPs than widthwise scaled models while scaling by both widthwise and depthwise models are in between. It is because of the model architecture and limited DoF of submodels. ResNets consist of convolution layers that have FLOPs of the parameters multiplied by feature sizes. The feature sizes get smaller as forwarding the layers and depthwise scaled submodels that omitted the latter layers make a model to require more FLOPs than widthwise scaled submodels that is scaled across all the layers.

It is worth noting that beyond the performance improvement (including that our proposed scaling method NeFL-W and NeFL-D over baselines in Table 6), *NeFL provides the more DoF for widthwise/depthwise scaling* that can be determined by the requirements of clients. It results in more clients to be participate in the FL pipeline. Also refer to Table 9 that has different scaling ratio $\gamma$. Note that in this case, FjORD (Horváth et al., 2021) outperforms NeFL-W. In this case with severe scaling factors (the worst model has 4% parameters of a global model), step sizes could not com-

Table 6: Ablation study by NeFL with five submodels for CIFAR-10 dataset under IID settings. We report Top-1 classification accuracies (%) for the worst-case submodel and the average of the performance of five submodels.

| Model | Method | Worst | Avg | Model | Method | Worst | Avg |
|---|---|---|---|---|---|---|---|
| ResNet18 | HeteroFL | 80.62 | 84.26 | ResNet34 | HeteroFL | 79.51 | 83.16 |
| | FjORD | 85.12 | 87.32 | | FjORD | 85.12 | 87.36 |
| | **NeFL-W** | **85.13** | **87.36** | | **NeFL-W** | **85.65** | **87.97** |
| | DepthFL | 64.80 | 82.44 | | DepthFL | 25.73 | 75.30 |
| | NeFL-D (N/L) | 86.29 | 88.12 | | NeFL-D (N/L) | 87.40 | 89.12 |
| | NeFL-$D_O$ (N/L) | 86.24 | 88.22 | | NeFL-$D_O$ (N/L) | 86.47 | 88.49 |
| | **NeFL-D** | **86.06** | **87.94** | | **NeFL-D** | **87.71** | **89.02** |
| | NeFL-$D_O$ | 85.98 | 88.20 | | NeFL-$D_O$ | 87.06 | 88.71 |
| | **NeFL-WD** | **86.86** | **87.88** | | **NeFL-WD** | **86.73** | **88.42** |
| | NeFL-WD (N/L) | 86.85 | 88.21 | | NeFL-WD (N/L) | 86.2 | 88.16 |

| Model | Method | Worst | Avg | Model | Method | Worst | Avg |
|---|---|---|---|---|---|---|---|
| Pre-trained ResNet18 | HeteroFL | 78.26 | 84.06 | Pre-trained ResNet34 | HeteroFL | 79.97 | 84.34 |
| | FjORD | 86.37 | 88.91 | | FjORD | 87.08 | 89.37 |
| | **NeFL-W** | **86.1** | **89.13** | | **NeFL-W** | **87.41** | **89.75** |
| | DepthFL | 47.76 | 82.85 | | DepthFL | 52.08 | 83.63 |
| | NeFL-D (N/L) | 86.95 | 89.77 | | NeFL-D (N/L) | 87.95 | 90.79 |
| | NeFL-$D_O$ (N/L) | 86.24 | 89.76 | | NeFL-$D_O$ (N/L) | 87.44 | 90.58 |
| | **NeFL-D** | **87.13** | **90.00** | | **NeFL-D** | **88.36** | **91.14** |
| | NeFL-$D_O$ | 87.02 | 89.72 | | NeFL-$D_O$ | 87.86 | 90.90 |
| | **NeFL-WD** | **88.61** | **89.60** | | **NeFL-WD** | **87.69** | **90.18** |
| | NeFL-WD (N/L) | 88.57 | 89.70 | | NeFL-WD (N/L) | 87.37 | 89.78 |

| Model | Method | Worst | Avg | Model | Method | Worst | Avg |
|---|---|---|---|---|---|---|---|
| ResNet56 | HeteroFL | 65.09 | 74.13 | ResNet110 | HeteroFL | 54.83 | 67.33 |
| | FjORD | 81.38 | 84.77 | | FjORD | 81.70 | 85.16 |
| | **NeFL-W** | **82.05** | **85.48** | | **NeFL-W** | **81.67** | **85.32** |
| | DepthFL | 72.94 | 86.19 | | DepthFL | 73.56 | 82.42 |
| | NeFL-D (N/L) | 84.38 | 86.13 | | NeFL-D (N/L) | 85.23 | 86.34 |
| | NeFL-$D_O$ (N/L) | 83.08 | 85.59 | | NeFL-$D_O$ (N/L) | 84 | 85.97 |
| | **NeFL-D** | **84.38** | **86.13** | | **NeFL-D** | **85.96** | **86.66** |
| | NeFL-$D_O$ | 81.97 | 85.37 | | NeFL-$D_O$ | 82.74 | 85.66 |
| | **NeFL-WD** | **83.92** | **86.00** | | **NeFL-WD** | **84.41** | **86.28** |
| | NeFL-WD (N/L) | 83.68 | 85.85 | | NeFL-WD (N/L) | 83.58 | 85.73 |

pensate the limited number of parameters and degraded the trainability with auxiliary parameters. However, NeFL-WD shows the best performance over other baselines that verify the well-balanced submodels show the better performance than ill-conditioned (too shallow or too narrow) submodels.

Table 7: Summarization of NeFL and baselines for ablation study

| | Depthwise scaling | Widthwise scaling | Adaptive step sizes |
|---|:---:|:---:|:---:|
| DepthFL | ✓ | | |
| FjORD, HeteroFL | | ✓ | |
| NeFL-D | ✓ | | ✓ |
| NeFL-W | | ✓ | ✓ |
| NeFL-WD | ✓ | ✓ | ✓ |

Table 8: Details of average FLOPs of submodels of $\gamma = [0.2, 0.4, 0.6, 0.8, 1]$

| Model | Metric | Method | | |
|---|---|:---:|:---:|:---:|
| | | Width/Depthwise scaling | Widthwise scaling | Depthwise scaling |
| ResNet18 | Param # | 6.71M | 6.71M | 6.68M |
| | FLOPs | 87.8M | 85M | 102M |
| ResNet34 | Param # | 12.6M | 12.8M | 12.9M |
| | FLOPs | 181M | 176M | 193M |
| ResNet56 | Param # | 0.51M | 0.52M | 0.51M |
| | FLOPs | 530M | 534M | 526M |
| ResNet110 | Param # | 1.05M | 1.06M | 1.04M |
| | FLOPs | 158M | 159M | 234M |

Table 9: Results of NeFL with five submodels ($\gamma = [0.04, 0.16, 0.36, 0.64, 1]$) for CIFAR-10 dataset on ResNet110. Results of NeFL with five submodels for CIFAR-10 dataset under IID settings. We report Top-1 classification accuracies (%) for the worst-case submodel and the average of the performance of five submodels.

| Model | Method | Model size | |
|---|---|:---:|:---:|
| | | Worst | Avg |
| ResNet110 | HeteroFL | 46.58 | 63.62 |
| | FjORD | 69.61 | 81.46 |
| | **NeFL-W** | 68.27 | 80.98 |
| | DepthFL | 11.00 | 53.91 |
| | **NeFL-D** | **75.4** | **84.31** |
| | **NeFL-WD** | **76.60** | **84.02** |

## B  EXPERIMENTAL DETAILS

### B.1  TRAINING DETAILS

The experiments in Table 1, Table 2, Table 4, Table 5 and Table 6 are evaluated by a total 500 communication rounds ($T$) with 100 clients ($M$). At each round, a fraction rate of 0.1 is used, indicating that 10 clients ($|\mathcal{C}_t| = 10$) transmit their weights to the server. During the training process of clients, local batch size of 32 and a local epoch of $E = 5$ are used. For training, we employ SGD optimizer (Ruder, 2016) without momentum and weight decay. The initial learning rate is set to 0.1 and decreases by a factor of $\frac{1}{10}$ at the halfway point and $\frac{3}{4}$ of the total communication rounds. The experiments in Table 3 are evaluated with the number of clients is $M = 10$, all of whom participate in the NeFL pipeline (with a fraction rate of 1). The experiment consists of $T = 100$ communication rounds, and each client performs local training for a single epoch ($E = 1$). We use a cosine annealing learning rate scheduling (Loshchilov & Hutter, 2017) with 500 steps of warmup and an initial learning rate 0.03. The input images are resized to a size of 256 and randomly cropped to a size of 224 with a padding size of 28. Note that utilizing layer normalization layers as consistent parameters, as opposed to BN layers that are inconsistent parameters, yields better performance.

### B.2  MODEL ARCHITECTURES

The ResNet18 architecture and ResNet34 architecture consist of four layers, while ResNet56 and ResNet110 have three layers. These layers are composed of blocks with different channel sizes, specifically (64, 128, 256, 512) for ResNet18/32 and (16, 32, 64) for ResNet56/110. Wide ResNet101_2 comprises four layers of bottleneck blocks with channel sizes (128, 256, 512, 1024) (He et al., 2016; Zagoruyko & Komodakis, 2016). The ViT-B/16 architecture consists of twelve layers, with each layer containing blocks comprising self-attention (SA) and feed-forward networks (FFN) (Dosovitskiy et al., 2021). The widthwise splitting for ViT models are implemented by varying the embedding dimension ($D$ in (Dosovitskiy et al., 2021)).

For the experiments presented in Table 1, Table 2, Table 4, Table 5 and Table 6, we consider five submodels with $\boldsymbol{\gamma} = [\gamma_1, \gamma_2, \gamma_3, \gamma_4, \gamma_5] = [0.2, 0.4, 0.6, 0.8, 1]$ and $\boldsymbol{\gamma} = [0.04, 0.16, 0.36, 0.64, 1]$ for Table 9. Additionally, for Table 3, we use three submodels with $\boldsymbol{\gamma} = [\gamma_1, \gamma_2, \gamma_3] = [0.5, 0.75, 1]$. Submodel details for ResNets and ViTs are detailed in Table 10 (ResNet18), Table 11 (ResNet34), Table 12 (ResNet56), Table 14 (ResNet110), Table 15 (Wide ResNet101_2) and Table 13 (ViT-B/16 In the tables, 1's and 0's denote the initial values of step sizes. A step size of zero indicates that a submodel does not include the corresponding block. Note that ResNets have a step size parameters for each block while ViTs have different step size parameters to be multiplied with SA and FFN. Submodels in NeFL-W are characterized by $\boldsymbol{\gamma}_D = [1, \ldots, 1]$ and $\boldsymbol{\gamma}_W$ with a target size, while submodels in NeFL-D are characterized by $\boldsymbol{\gamma}_W = [1, \ldots, 1]$ and $\boldsymbol{\gamma}_D$ with a target size. Submodels in NeFL-WD are characterized by target size $\boldsymbol{\gamma}_W \boldsymbol{\gamma}_D$. Corresponding number of parameters and FLOPs are provided in Table 8.

### B.3  EXAMPLE ON PARAMETER AVERAGING

Consider an example with five submodels and suppose that a convolutional layer from the first block is included in submodel 1, 3, and 5. Assume that submodel 1 and submodel 5 are trained twice (two clients), while submodel 3 is trained three times at a communication round. Then, we have $|\mathcal{M}_2| = |\mathcal{M}_4| = 0, |\mathcal{M}_1| = |\mathcal{M}_5| = 2$, and $|\mathcal{M}_3| = 3$. Now, delving into the parameter averaging process, the parameters exclusive to submodel 5 ($\phi_{1,5} \setminus \phi_{1,3}$) are averaged using two updated weights ($\mathcal{M}_5$). Likewise, the parameters possessed by submodel 3 but not by submodel 1 ($\phi_{1,3} \setminus \phi_{1,1}$) are averaged using five weights ($\mathcal{M}_5 \cup \mathcal{M}_3$). Finally, the parameters of submodel 1 $\phi_{1,1}$, that is trained seven times, are averaged using seven weights ($\mathcal{M}_5 \cup \mathcal{M}_3 \cup \mathcal{M}_1$). This approach ensures that consistent parameters are appropriately averaged, taking into account their depthwise inclusion and the widthwise number of occurrences across different submodels.

### B.4  DATASET

**CIFAR10/100.** The CIFAR10 dataset consists of 60000 images (train dataset consists of 50000 samples and test dataset consists of 10000 samples). $32 \times 32 \times 3$ color images are categorized by

10 classes, with 6000 images per class (Krizhevsky et al.). For FL with each client has 500 data samples for $M = 100$, and 5000 data samples for $M = 10$. We perform data augmentation and pre-processing of random cropping (by $32 \times 32$ with padding of 4), random horizontal flip, and normalization by mean of $(0.4914, 0.4822, 0.4465)$ and standard deviation of $(0.2023, 0.1994, 0.2010)$.

**CINIC10.** The CIFAR10 dataset consists of 270000 $32 \times 32 \times 3$ color images (train dataset consists of 90000 samples and validation and test dataset consists of 90000 samples respectively) in 10 classes (Darlow et al., 2018). It is constructed from ImageNet and CIFAR10. We perform data augmentation and pre-processing of random cropping (by $32 \times 32$ with padding of 4), random horizontal flip, and normalization by mean of $(0.47889522, 0.47227842, 0.43047404)$ and standard deviation of $(0.24205776, 0.23828046, 0.25874835)$.

**SVHN.** The SVHN dataset consists of 73257 digits for training and 26032 digits for testing of $32 \times 32 \times 3$ color images (Netzer et al., 2011). The deataset is obtained from house numbers in Google Stree view images in 10 classes (digit '0' to digit '9'). For FL for $M = 100$ each client has 732 samples. We perform data augmentation and pre-processing of random cropping (by $32 \times 32$ with padding of 2), color jitter (by brightness of 63/255, saturation=[0.5, 1.5] and contrast=[0.2, 1.8] implmented by `torchvision.transforms.ColorJitter`), and normalization by mean of $(0.4376821, 0.4437697, 0.47280442)$ and standard deviation of $(0.19803012, 0.20101562, 0.19703614)$.

### B.5 BASELINES

**HeteroFL & FjORD.** HeteroFL (Diao et al., 2021) and FjORD (Horváth et al., 2021) are width-wise splitting methods designed to address the challenges posed by client heterogeneity in FL. While both methods aim to mitigate the impact of heterogeneity, these are several key differences between them. Firstly, HeteroFL does not utilize separate (i.e., inconsistent) parameters for BN layers in its submodels, whereas FjROD incorporates distinct BN layer for each submodel. This difference in handling BN layers can impact the learning dynamics and model performance under IID settings. Secondly, HeteroFL employs static batch normalization, where BN statistics are updated using the entire dataset after the training process. On the other hand, FjORD updates BN statistics during training. Lastly, HeteroFL utilizes a masked cross-entropy loss to address the statistical heterogeneity among clients. This loss function helps to mitigate the impact of clients with the statistical heterogeneity. In our implementation of HeteroFL, the masked cross-entropy loss is not utilized.

**DepthFL.** The model is split depthwise, and an auxiliary bottleneck layer is included as an independent classifier. We implement DepthFL (Kim et al., 2023) without separate bottleneck layers for fair comparison without additional parameters. Then, DepthFL is a special case of NeFL-D without inconsistent parameters. Furthermore, due to the accuracy degradation, we omitted knowledge distillation noted in (Kim et al., 2023). Instead, our DepthFL models incorporate downsampling layers that adjust the feature size to match the input sizes of the classifier. It is important to note that the auxiliary bottleneck layers for submodels in DepthFL can be interpreted as parameter decoupling, as discussed in Section 4.2.

### B.6 PRE-TRAINED MODELS

The pre-trained models on Table 2 and Table 3 are trained on ImageNet-1k (Deng et al., 2009) as following recipes (Pyt, 2023):

**ResNet18/34.** The models are trained by epochs of 90, batch size of 32, SGD optimizer (Ruder, 2016), learning rate of 0.1 with momentum of 0.9 and weight decay of 0.0001, where learning rate is decreased by a factor of 0.1 every 30 epochs.

**Wide ResNet101_2.** The model is trained by epochs of 90, batch size of 32, SGD optimizer (Ruder, 2016), learning rate of 0.1 with momentum of 0.9 and weight decay of 0.0001, where the learning scheduler is cosine learning rate (Loshchilov & Hutter, 2017) and warming up restarts for 256 epochs.

**ViT-B/16.** The model is trained by epochs of 300, batch size of 512, AdamW optimizer (Loshchilov & Hutter, 2019) with learning rate of 0.003 and weight decay of 0.3. The learning scheduler is cosine annealing (Loshchilov & Hutter, 2017) after linear warmup method with decay of 0.033 for 30 epochs. Additionally, the random augmentation (Cubuk et al., 2020), random mixup with $\alpha = 0.2$ (Zhang et al., 2018), cutmix of $\alpha = 1$ (Yun et al., 2019), repeated augmentation, label smoothing of 0.11 (Szegedy et al., 2016), clipping gradient norm to 1, model exponential moving average (EMA) are employed.

### B.7 DYNAMIC ENVIRONMENT

We simulate a dynamic environment by randomly selecting which submodel to be trained by each client during every communication round. In our experiments for Table 1, Table 2, Table 4, Table 5 and Table 6, we have an equal number of five tiers of clients ($M/N_s = 20$ for all tiers of clients). The resource-constrained clients (tier 1) randomly select models between $\gamma = 0.2, 0.4, 0.6$, clients in tier 2 randomly select models from the set $\gamma = 0.2, 0.4, 0.6, 0.8$, clients in tier 3 randomly select models from the set $\gamma = 0.2, 0.4, 0.6, 0.8, 1$, clients in tier 4 randomly select models from the set $\gamma = 0.4, 0.6, 0.8, 1$, and the resource-richest clients (tier 5) randomly select models from the set $\gamma = 0.6, 0.8, 1$. In our experiments for Table 3 involving three submodels and 10 clients, the tier 1 clients (3 out of 10 total clients) select $\gamma = 0.5$, tier 2 clients (3 out of 10 total clients) select $\gamma = 0.75$ and tier 3 clients (4 out of 10 total clients) select $\gamma = 1$. By allowing clients to randomly choose from the available submodels, our setup reflects the dynamic nature in which clients may encounter communication computing bottlenecks during each iteration.

### B.8 PSEUDOCODE FOR PARAMETER AVERAGING

```python
import numpy as np

M = [[] for _ in range(Ns)]
for i in range(len(uploaded_weights)):
    for k in range(Ns):
        if uploaded_weights[i]==theta[k]: # parameters of submodel k
            M[k].append(uploaded_weights[i])

def NeFedAvg(M): # consistent parameters
    for block in theta_c[-1]: # global model parameters
        for key in block: # depthwise access by block by block
        num_submodel_uploaded=[], submodel_idx=[], gamma_W_block=[0]
            for k in range(Ns):
                if key in theta_c[k]:
                    submodel_idx.append(k)
                    num_submodels.append(len(M[k]))
                    card = np.cumsum(num_submodels[::-1])[::-1] # cardinality
                    gamma_W_block.append(gamma_W[k])
            for i in range(len(submodel_idx)): # widthwise access
                start = math.ceil(param_size*gamma_W_block[i])
                end = math.ceil(param_size*gamma_W_block[i+1])
                for w in M[submodel_idx[i]]:
                    theta_c_avg[key][start:end]+=w[key][start:end]/card[i]
    return theta_c_avg

def InconsistentParamAvg(M): # inconsistent parameters
    for k in range(Ns):
        for key in theta_ic[k]:
            for w in M[k]:
            theta_ic_avg[k][key] += w[key]
    return theta_ic_avg
```

Table 10: Details of $\gamma$ of NeFL-D and NeFL-WD on ResNet18

| Model index | Model size $\gamma$ | $\gamma_W$ | $\gamma_D$ | NeFL-D (ResNet18) | | | |
|---|---|---|---|---|---|---|---|
| | | | | Layer 1 (64) | Layer 2 (128) | Layer3 (256) | Layer 4 (512) |
| 1 | 0.20 | 1 | 0.20 | 1,1 | 0,0 | 1,1 | 0,0 |
| 2 | 0.38 | 1 | 0.38 | 1,0 | 0,0 | 1,0 | 1,0 |
| 3 | 0.57 | 1 | 0.57 | 1,1 | 1,1 | 1,1 | 1,0 |
| 4 | 0.81 | 1 | 0.81 | 1,0 | 1,1 | 0,0 | 1,1 |
| 5 | 1 | 1 | 1 | 1,1 | 1,1 | 1, 1 | 1,1 |
| Model index | Model size $\gamma$ | $\gamma_W$ | $\gamma_D$ | NeFL-WD (ResNet18) | | | |
| | | | | Layer 1 (64) | Layer 2 (128) | Layer3 (256) | Layer 4 (512) |
| 1 | 0.20 | 0.34 | 0.58 | 1,1 | 1,1 | 1, 1 | 1,0 |
| 2 | 0.4 | 0.4 | 1 | 1,1 | 1,1 | 1, 1 | 1,1 |
| 3 | 0.6 | 0.6 | 1 | 1,1 | 1,1 | 1, 1 | 1,1 |
| 4 | 0.8 | 0.8 | 1 | 1,1 | 1,1 | 1, 1 | 1,1 |
| 5 | 1 | 1 | 1 | 1,1 | 1,1 | 1, 1 | 1,1 |

Table 11: Details of $\gamma$ of NeFL-D and NeFL-WD on ResNet34

| Model index | Model size $\gamma$ | $\gamma_W$ | $\gamma_D$ | NeFL-D (ResNet34) | | | |
|---|---|---|---|---|---|---|---|
| | | | | Layer 1 (64) | Layer 2 (128) | Layer3 (256) | Layer 4 (512) |
| 1 | 0.23 | 1 | 0.23 | 1,0,0 | 1,0,0,0 | 1,0,0,0,0,0 | 1,0,0 |
| 2 | 0.39 | 1 | 0.39 | 1,1,1 | 1,1,1,1 | 1,1,0,0,0,1 | 1,0,0 |
| 3 | 0.61 | 1 | 0.61 | 1,1,1 | 1,1,1,1 | 1,1,0,0,0,1 | 1,0,1 |
| 4 | 0.81 | 1 | 0.81 | 1,1,1 | 1,0,0,1 | 1,1,0,0,0,1 | 1,1,1 |
| 5 | 1 | 1 | 1 | 1,1,1 | 1,1,1,1 | 1,1,1,1,1,1 | 1,1,1 |
| Model index | Model size $\gamma$ | $\gamma_W$ | $\gamma_D$ | NeFL-WD (ResNet34) | | | |
| | | | | Layer 1 (64) | Layer 2 (128) | Layer3 (256) | Layer 4 (512) |
| 1 | 0.20 | 0.38 | 0.53 | 1,1,1 | 1,0,0,1 | 1,0,0,0,0,1 | 1,0,1 |
| 2 | 0.40 | 0.63 | 0.64 | 1,1,1 | 1,0,0,1 | 1,1,1,0,0,1 | 1,0,1 |
| 3 | 0.60 | 0.77 | 0.78 | 1,1,1 | 1,1,1,1 | 1,1,1,1,0,1 | 1,0,1 |
| 4 | 0.80 | 0.90 | 0.89 | 1,1,1 | 1,1,1,1 | 1,1,1,0,0,1 | 1,1,1 |
| 5 | 1 | 1 | 1 | 1,1,1 | 1,1,1,1 | 1,1,1,1,1,1 | 1,1,1 |

Table 12: Details of $\gamma$ of NeFL-D and NeFL-WD on ResNet56

**NeFL-D (ResNet56)**

| Model index | Model size $\gamma$ | $\gamma_W$ | $\gamma_D$ | Layer 1 (16) | Layer 2 (32) | Layer3 (64) |
|---|---|---|---|---|---|---|
| 1 | 0.2 | 1 | 0.2 | 1, 1, 0, 0, 0, 0, 0, 0, 0 | 1, 1, 0, 0, 0, 0, 0, 0, 0 | 1, 1, 0, 0, 0, 0, 0, 0, 0 |
| 2 | 0.4 | 1 | 0.4 | 1, 1, 1, 0, 0, 0, 0, 0, 0 | 1, 1, 1, 0, 0, 0, 0, 0, 0 | 1, 1, 1, 0, 0, 0, 0, 0, 0 |
| 3 | 0.6 | 1 | 0.6 | 1, 1, 1, 1, 0, 0, 0, 0, 0 | 1, 1, 1, 1, 0, 0, 0, 0, 0 | 1, 1, 1, 1, 1, 0, 0, 0, 0 |
| 4 | 0.8 | 1 | 0.8 | 1, 1, 1, 1, 1, 1, 1, 1, 1 | 1, 1, 1, 1, 1, 1, 1, 1, 0 | 1, 1, 1, 1, 1, 1, 1, 0, 0 |
| 5 | 1 | 1 | 1 | 1, 1, 1, 1, 1, 1, 1, 1, 1 | 1, 1, 1, 1, 1, 1, 1, 1, 1 | 1, 1, 1, 1, 1, 1, 1, 1, 1 |

**NeFL-D$_O$ (ResNet56)**

| Model index | Model size $\gamma$ | $\gamma_W$ | $\gamma_D$ | Layer 1 (16) | Layer 2 (32) | Layer3 (64) |
|---|---|---|---|---|---|---|
| 1 | 0.2 | 1 | 0.2 | 1, 8, 0, 0, 0, 0, 0, 0, 0 | 1, 8, 0, 0, 0, 0, 0, 0, 0 | 1, 8, 0, 0, 0, 0, 0, 0, 0 |
| 2 | 0.4 | 1 | 0.4 | 1, 1, 7, 0, 0, 0, 0, 0, 0 | 1, 1, 7, 0, 0, 0, 0, 0, 0 | 1, 1, 6, 0, 0, 0, 0, 0, 0 |
| 3 | 0.6 | 1 | 0.6 | 1, 1, 1, 6, 0, 0, 0, 0, 0 | 1, 1, 1, 6, 0, 0, 0, 0, 0 | 1, 1, 1, 4, 0, 0, 0, 0 |
| 4 | 0.8 | 1 | 0.8 | 1, 1, 1, 1, 1, 1, 1, 1, 1 | 1, 1, 1, 1, 1, 1, 2, 0 | 1, 1, 1, 1, 1, 3, 0, 0 |
| 5 | 1 | 1 | 1 | 1, 1, 1, 1, 1, 1, 1, 1, 1 | 1, 1, 1, 1, 1, 1, 1, 1, 1 | 1, 1, 1, 1, 1, 1, 1, 1, 1 |

**NeFL-WD (ResNet56)**

| Model index | Model size $\gamma$ | $\gamma_W$ | $\gamma_D$ | Layer 1 (16) | Layer 2 (32) | Layer3 (64) |
|---|---|---|---|---|---|---|
| 1 | 0.2 | 0.46 | 0.43 | 1, 1, 1, 0, 0, 0, 0, 0 | 1, 1, 1, 0, 0, 0, 0, 0 | 1, 1, 1, 0, 0, 0, 0, 0 |
| 2 | 0.4 | 0.61 | 0.66 | 1, 1, 1, 1, 1, 1, 0, 0 | 1, 1, 1, 1, 1, 0, 0, 0 | 1, 1, 1, 1, 1, 0, 0, 0 |
| 3 | 0.6 | 0.77 | 0.77 | 1, 1, 1, 1, 1, 1, 1, 0, 0 | 1, 1, 1, 1, 1, 1, 0, 0 | 1, 1, 1, 1, 1, 1, 0, 0 |
| 4 | 0.8 | 0.90 | 89 | 1, 1, 1, 1, 1, 1, 1, 1, 0 | 1, 1, 1, 1, 1, 1, 1, 0 | 1, 1, 1, 1, 1, 1, 1, 0 |
| 5 | 1 | 1 | 1 | 1, 1, 1, 1, 1, 1, 1, 1, 1 | 1, 1, 1, 1, 1, 1, 1, 1 | 1, 1, 1, 1, 1, 1, 1, 1 |

Table 13: Details of $\gamma$ of NeFL-D and NeFL-W on ViT-B/16

**NeFL-D (ViT-B/16)**

| Model index | Model size $\gamma$ | $\gamma_W$ | $\gamma_D$ | Block |
|---|---|---|---|---|
| 1 | 0.5 | 1 | 0.50 | 1,1,1,1,1,1,0,0,0,0,0,0 |
| 2 | 0.75 | 1 | 0.75 | 1,1,1,1,1,1,1,1,1,0,0,0 |
| 3 | 1 | 1 | 1 | 1,1,1,1,1,1,1,1,1,1,1,1 |

**NeFL-W (ViT-B/16)**

| Model index | Model size $\gamma$ | $\gamma_W$ | $\gamma_D$ | Block |
|---|---|---|---|---|
| 1 | 0.5 | 0.5 | 1 | 1,1,1,1,1,1,1,1,1,1,1,1 |
| 2 | 0.75 | 0.75 | 1 | 1,1,1,1,1,1,1,1,1,1,1,1 |
| 3 | 1 | 1 | 1 | 1,1,1,1,1,1,1,1,1,1,1,1 |

Table 14: Details of γ of NeFL on ResNet110

**NeFL-D (ResNet110)**

| Model size | | | Layer 1 (16) | Layer 2 (32) | Layer3 (64) |
|---|---|---|---|---|---|
| γ | γ_W | γ_D | | | |
| 0.2 | 1 | 0.20 | 1, 1, 1, 1, 1, 1, 1, 1, 1, 1, 1, 1, 1, 1, 0, 0 | 1, 1, 1, 0, 0, 0, 0, 0, 0, 0, 0, 0, 0, 0, 0, 0, 0, 0 | 1, 1, 0, 0, 0, 0, 0, 0, 0, 0, 0, 0, 0, 0, 0, 0, 0, 0 |
| 0.4 | 1 | 0.40 | 1, 1, 1, 1, 1, 1, 1, 1, 1, 1, 1, 0, 0, 0 | 1, 1, 1, 1, 0, 0, 0, 0, 0, 0, 0, 0, 0, 0, 0, 0, 0, 0 | 1, 1, 1, 1, 0, 0, 0, 0, 0, 0, 0, 0, 0, 0, 0, 0, 0, 0 |
| 0.6 | 1 | 0.60 | 1, 1, 1, 1, 1, 1, 1, 1, 1, 1, 1, 1, 1, 1, 1, 0 | 1, 1, 1, 1, 1, 1, 1, 1, 0, 0, 0, 0, 0, 0, 0, 0, 0, 0 | 1, 1, 1, 1, 1, 1, 0, 0, 0, 0, 0, 0, 0, 0, 0, 0, 0, 0 |
| 0.8 | 1 | 0.80 | 1, 1, 1, 1, 1, 1, 1, 1, 1, 1, 1, 1, 1, 1, 1, 0, 0 | 1, 1, 1, 1, 1, 1, 1, 1, 1, 1, 0, 0, 0, 0, 0, 0, 0, 0 | 1, 1, 1, 1, 1, 1, 1, 1, 1, 0, 0, 0, 0, 0, 0, 0, 0, 0 |
| 1 | 1 | 1 | 1, 1, 1, 1, 1, 1, 1, 1, 1, 1, 1, 1, 1, 1, 1, 1, 1 | 1, 1, 1, 1, 1, 1, 1, 1, 1, 1, 1, 1, 1, 1, 1, 1, 1, 1 | 1, 1, 1, 1, 1, 1, 1, 1, 1, 1, 1, 1, 1, 1, 1, 1, 1, 1 |

**NeFL-WD (ResNet110)**

| Model size | | | Layer 1 (16) | Layer 2 (32) | Layer3 (64) |
|---|---|---|---|---|---|
| γ | γ_W | γ_D | | | |
| 0.2 | 0.46 | 0.44 | 1, 1, 1, 1, 1, 0, 0, 0, 0, 0, 0, 0, 0, 0, 0, 1 | 1, 1, 1, 1, 0, 0, 0, 0, 0, 0, 0, 0, 0, 0, 0, 0, 0, 1 | 1, 1, 1, 1, 0, 0, 0, 0, 0, 0, 0, 0, 0, 0, 0, 0, 0, 1 |
| 0.4 | 0.60 | 0.66 | 1, 1, 1, 1, 1, 1, 1, 0, 0, 0, 0, 0, 0, 0, 0, 1 | 1, 1, 1, 1, 1, 1, 1, 0, 0, 0, 0, 0, 0, 0, 0, 0, 0, 1 | 1, 1, 1, 1, 1, 1, 1, 0, 0, 0, 0, 0, 0, 0, 0, 0, 0, 1 |
| 0.6 | 0.77 | 0.77 | 1, 1, 1, 1, 1, 1, 1, 1, 1, 0, 0, 0, 0, 0, 0, 1 | 1, 1, 1, 1, 1, 1, 1, 1, 1, 1, 0, 0, 0, 0, 0, 0, 0, 1 | 1, 1, 1, 1, 1, 1, 1, 1, 1, 0, 0, 0, 0, 0, 0, 0, 0, 1 |
| 0.8 | 0.90 | 0.89 | 1, 1, 1, 1, 1, 1, 1, 1, 1, 1, 1, 0, 0, 0, 1, 1 | 1, 1, 1, 1, 1, 1, 1, 1, 1, 1, 1, 1, 0, 0, 0, 0, 1, 1 | 1, 1, 1, 1, 1, 1, 1, 1, 1, 1, 1, 1, 1, 0, 0, 1, 1, 1 |
| 1 | 1 | 1 | 1, 1, 1, 1, 1, 1, 1, 1, 1, 1, 1, 1, 1, 1, 1, 1 | 1, 1, 1, 1, 1, 1, 1, 1, 1, 1, 1, 1, 1, 1, 1, 1, 1, 1 | 1, 1, 1, 1, 1, 1, 1, 1, 1, 1, 1, 1, 1, 1, 1, 1, 1, 1 |

**NeFL-D (ResNet110)**

| Model size | | | Layer 1 (16) | Layer 2 (32) | Layer3 (64) |
|---|---|---|---|---|---|
| γ | γ_W | γ_D | | | |
| 0.04 | 1 | 0.04 | 1, 0, 0, 0, 0, 0, 0, 0, 0, 0, 0, 0, 0, 0, 0, 0 | 1, 0, 0, 0, 0, 0, 0, 0, 0, 0, 0, 0, 0, 0, 0, 0, 0, 0 | 1, 0, 0, 0, 0, 0, 0, 0, 0, 0, 0, 0, 0, 0, 0, 0, 0, 0 |
| 0.16 | 1 | 0.16 | 1, 1, 1, 0, 0, 0, 0, 0, 0, 0, 0, 0, 0, 0, 0, 0 | 1, 1, 1, 0, 0, 0, 0, 0, 0, 0, 0, 0, 0, 0, 0, 0, 0, 0 | 1, 1, 0, 0, 0, 0, 0, 0, 0, 0, 0, 0, 0, 0, 0, 0, 0, 0 |
| 0.36 | 1 | 0.37 | 1, 1, 1, 1, 1, 0, 0, 0, 0, 0, 0, 0, 0, 0, 0, 0 | 1, 1, 1, 1, 1, 0, 0, 0, 0, 0, 0, 0, 0, 0, 0, 0, 0, 0 | 1, 1, 1, 1, 1, 0, 0, 0, 0, 0, 0, 0, 0, 0, 0, 0, 0, 0 |
| 0.64 | 1 | 0.65 | 1, 1, 1, 1, 1, 1, 1, 1, 1, 0, 0, 0, 0, 0, 0, 0 | 1, 1, 1, 1, 1, 1, 1, 1, 1, 0, 0, 0, 0, 0, 0, 0, 0, 0 | 1, 1, 1, 1, 1, 1, 1, 1, 1, 0, 0, 0, 0, 0, 0, 0, 0, 0 |
| 1 | 1 | 1 | 1, 1, 1, 1, 1, 1, 1, 1, 1, 1, 1, 1, 1, 1, 1, 1 | 1, 1, 1, 1, 1, 1, 1, 1, 1, 1, 1, 1, 1, 1, 1, 1, 1, 1 | 1, 1, 1, 1, 1, 1, 1, 1, 1, 1, 1, 1, 1, 1, 1, 1, 1, 1 |

**NeFL-WD (ResNet110)**

| Model size | | | Layer 1 (16) | Layer 2 (32) | Layer3 (64) |
|---|---|---|---|---|---|
| γ | γ_W | γ_D | | | |
| 0.04 | 0.26 | 0.16 | 1, 1, 1, 0, 0, 0, 0, 0, 0, 0, 0, 0, 0, 0, 0, 0 | 1, 1, 1, 0, 0, 0, 0, 0, 0, 0, 0, 0, 0, 0, 0, 0, 0, 0 | 1, 1, 1, 0, 0, 0, 0, 0, 0, 0, 0, 0, 0, 0, 0, 0, 0, 0 |
| 0.16 | 0.42 | 0.38 | 1, 1, 1, 1, 1, 0, 0, 0, 0, 0, 0, 0, 0, 0, 0, 0 | 1, 1, 1, 1, 1, 0, 0, 0, 0, 0, 0, 0, 0, 0, 0, 0, 0, 0 | 1, 1, 1, 1, 1, 0, 0, 0, 0, 0, 0, 0, 0, 0, 0, 0, 0, 0 |
| 0.36 | 0.59 | 0.61 | 1, 1, 1, 1, 1, 1, 1, 0, 0, 0, 0, 0, 0, 0, 0, 0 | 1, 1, 1, 1, 1, 1, 1, 0, 0, 0, 0, 0, 0, 0, 0, 0, 0, 0 | 1, 1, 1, 1, 1, 1, 1, 0, 0, 0, 0, 0, 0, 0, 0, 0, 0, 0 |
| 0.64 | 0.77 | 0.83 | 1, 1, 1, 1, 1, 1, 1, 1, 1, 1, 1, 0, 0, 0, 0, 0 | 1, 1, 1, 1, 1, 1, 1, 1, 1, 1, 1, 0, 0, 0, 0, 0, 0, 0 | 1, 1, 1, 1, 1, 1, 1, 1, 1, 1, 1, 0, 0, 1, 1, 1, 1, 1 |
| 1 | 1 | 1 | 1, 1, 1, 1, 1, 1, 1, 1, 1, 1, 1, 1, 1, 1, 1, 1 | 1, 1, 1, 1, 1, 1, 1, 1, 1, 1, 1, 1, 1, 1, 1, 1, 1, 1 | 1, 1, 1, 1, 1, 1, 1, 1, 1, 1, 1, 1, 1, 1, 1, 1, 1, 1 |

Table 15: Details of γ of NeFL on Wide ResNet101_2

**NeFL-D (Wide ResNet101_2)**

| Model size | | | Layer 1 (128) | Layer 2 (256) | Layer3 (512) | Layer 4 (1024) |
|---|---|---|---|---|---|---|
| γ | γ_W | γ_D | | | | |
| 0.5 | 1 | 0.51 | 1,1,1 | 1,1,1 | 1, 1, 1, 1, 1, 1, 1, 1, 1, 1, 0, 0, 0, 0, 0, 0, 0, 0, 0, 0, 0, 0, 0 | 1, 1, 0 |
| 0.75 | 1 | 0.75 | 1,1,1 | 1,1,1 | 1, 1, 1, 1, 1, 1, 1, 1, 1, 1, 1, 1, 1, 1, 1, 0, 0, 0, 0, 0, 0, 0, 0 | 1, 1, 1 |
| 1 | 1 | 1 | 1,1,1 | 1,1,1 | 1, 1, 1, 1, 1, 1, 1, 1, 1, 1, 1, 1, 1, 1, 1, 1, 1, 1, 1, 1, 1, 1, 1 | 1, 1, 1 |

## C  INTERPRETING SCALING WITH ODE SOLVER

We present a toy example of ODE solvers in Figure 4. The black line representing an actual function is $0.1t + \sin(0.2t) + \cos(0.3t)$ and the red line representing a discretized approximation of the actual function (i.e., a full neural network). Here, it was implemented by ODE solver of step size $h = 2$. The green line representing a widthwise scaled submodel has numerical errors on each step. The blue solid line is a depthwise scaled submodel with step size $h = 3$. The blue dashed line has forward with same steps, but with optimized step sizes. It denotes that the optimized step sizes can decrease the numerical error. The cyan colored line is implemented by improved Euler method (Hairer et al., 2000) with step size $h = 3$. It denotes that even with same steps with blue solid lines, and optimizing $dy/dt$ also contributes to decrease the numerical error. The figure shows a toy example why each submodel can still work well with less parameters.

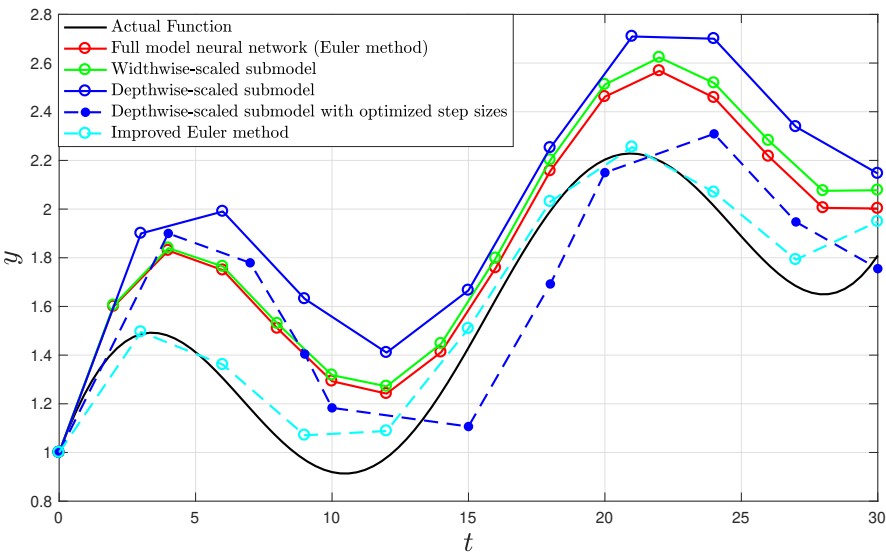

Figure 4: Example of widthwise/depthwise model scaling by ODE solver.