# OpenReview forum: "NeFL: Nested Federated Learning for Heterogeneous Clients"
_ICLR.cc/2024/Conference — Submitted to ICLR 2024_

### Official Review · Reviewer_7d7r · 2023-10-29

**Soundness:** 2 fair
**Presentation:** 3 good
**Contribution:** 2 fair
**Rating:** 5
**Confidence:** 4

**Summary:**

This paper proposes a new sub-model training method in federated settings.
The proposed method, NeFL, creates sub-models for participating clients by reducing both the depth and width of a full model.
The way of creating sub-models is inspired by the interpretation that a model with skip-connections can be regarded as ordinary differential equations (ODEs).
Depending on each client's resource, the client picks the feasible sub-model that can be trained locally.
Therefore, training costs are reduced.

**Strengths:**

This paper targets a practical and important problem of reducing training costs in FL with large models, especially for large models. Several contributions are highlighted below.

**1**, The motivation for reducing both depth and width is interesting.

In Sec.3, the authors justify the way of creating sub-models in NeFL. Inspired by the interpretation, that models with skip connections can be seen as ODEs, the authors argue that it is reasonable to remove some residual blocks, as skipping some steps in solving ODEs.
The motivation is valid and interesting.

**2**, Experiments cover different sub-model methods.

The author compared NeFL with both depth scaling and width scaling methods. The results show promising results.

**Weaknesses:**

**1**, Lack of explanations why NeFL is better than scaling width or depth only.

While the authors show NeFL results in higher accuracy compared to width-scaling and depth-scaling methods, necessary explanations/intuitions are missing.
As a reader, I do not see which part of NeFL contributes to performance gains.
I believe the authors need to provide more insights to demystify NeFL. For instance, why scaling both width and depth is better than scaling one dimension only?


**2**, Lack of contributions.

Essentially, NeFL is a combination of width-scaling (e.g., HeteroFL) and depth-scaling (e.g., DepthFL). Other than that, I did not see a nontrivial contribution in this work.
Importantly, it is doubtful such a combination can bring significant improvement over prior works.

**Questions:**

None

---

> ### Author Response · Authors · 2023-11-14
> **Response to Reviewer 7d7r**
>
> Dear Reviewer 7d7r,
>
> We sincerely thank you for your careful review and valuable feedback. We address your comments and questions below. In the revised draft, we mark our revision in $\color{blue}\text{blue}$.
>
> ---
> **1**. Why does NeFL outperform width or depth only scaling?
>
> **Width/depth-balanced models have performance gains over one dimension scaled models with the same number of parameters.**
>
> It has been shown in centralized learning that both width and depth dimension scaled models outperform one-dimension-scaled models with the same number of parameters (Tan & Le, 2019). It is also important for FL that scales a global model.
>
> However, in FL that scales a global model into submodels by one dimension (Horvath et al., 2021; Diao et al., 2021; Kim et al., 2023), the performance of the worst submodel (with the least number of parameters) degrades severely. Therefore, our proposed well-balanced scaling method improves the performance of the worst submodel.
> - We added the explanations in “Section 2 model splitting”.
>
> > Mingxing Tan and Quoc Le. EfficientNet: Rethinking model scaling for convolutional neural networks. In ICML, 2019.
>
> > Samuel Horvath, Stefanos Laskaridis, Mario Almeida, Ilias Leontiadis, Stylianos Venieris, and Nicholas Lane. FjORD: Fair and accurate federated learning under heterogeneous targets with ordered dropout. In NeurIPS, 2021.
>
> > Enmao Diao, Jie Ding, and Vahid Tarokh. HeteroFL: Computation and communication efficient federated learning for heterogeneous clients. In ICLR, 2021.
>
> > Minjae Kim, Sangyoon Yu, Suhyun Kim, and Soo-Mook Moon. DepthFL : Depthwise federated learning for heterogeneous clients. In ICLR, 2023.
>
> **Inconsistent parameters also contribute to performance gain.**
>
> We introduce inconsistent parameters to compensate for aggregating parameters of submodels of different model architecture. It includes batch normalization layers and learnable step size parameters motivated by ODE solver. Table 6 in Appendix A.3 provides the performance gain of inconsistent parameters. The insights on inconsistent parameters can be observed in Figure 1 and Figure 4. The inconsistent parameters can compensate for numerical error caused by skipping a few blocks (depthwise scaling) or widthwise scaling.
>
> - We clarified the explanations in “Section 4.2 Inconsistency”.
> ---
> **2**. Contributions beyond scaling
>
> **Inconsistent parameters and ParameterAveraging method.**
>
> In addition to both widthwise and depthwise scaling, NeFL introduces inconsistent parameters. The effectiveness of inconsistent parameters are presented in ablation study (Table 6 in Appendix A.3). Comparison of NeFL-W and HeteroFL and comparison of NeFL-D and DepthFL provides the performance gain of the inconsistent parameters.
> Moreover, we propose a simple yet novel parameter averaging method (Algorithm 2) that corresponds to general scaling methods we presented where the submodels consist of consistent and inconsistent parameters.
>
> **Exploration in line with recent studies of federated learning.**
>
> We explore the NeFL aligns with recent studies on FL (Chen et al., 2023; Qu et al., 2022). It is notable that a pre-trained model that is not trained in a nested manner with several submodels has performance gains with NeFL compared to previous works. We also observe that ViTs on NeFL are effective in non-IID settings compared to ResNet with more parameters.
>
> > Hong-You Chen, Cheng-Hao Tu, Ziwei Li, Han Wei Shen, and Wei-Lun Chao. On the importance and applicability of pre-training for federated learning. In ICLR, 2023.
>
> > Liangqiong Qu, Yuyin Zhou, Paul Pu Liang, Yingda Xia, Feifei Wang, Ehsan Adeli, Li Fei-Fei, and Daniel Rubin. Rethinking architecture design for tackling data heterogeneity in federated learning. In CVPR, 2022.
>
> **Comprehensive empirical results**
>
> Our paper has comprehensive empirical results. The results compare a variety of recent frameworks and present the effectiveness of NeFL.
> - We clarified the contributions in “Section 1”.
>
> ---
> Thank you again for your valuable time and effort spent reviewing!

---

### Official Review · Reviewer_HZdf · 2023-11-01

**Soundness:** 3 good
**Presentation:** 2 fair
**Contribution:** 3 good
**Rating:** 6
**Confidence:** 3

**Summary:**

The authors introduce a method called Nested FL which helps to improve performance when training with heterogeneous clients.

**Strengths:**

The authors do a good job describing the state of the art, the work is timely as the clients that are participating in FL are becoming increasingly diverse. Thus efficient schemes of tackling and even exploiting that heterogeneity are highly desirable. In principle, I found the concept of averaging nesting consistent parameters is simple yet novel and interesting. Finally the empirical performance based on the empirical numbers provided is encouraging. Appreciated that they have compared against a variety of recent frameworks that aim to do a similar task so we can get a broad idea of how the scheme performs.

**Weaknesses:**

- Lack of clarity what happens with stragglers and/or dropped nodes in the proposed framework.
- Reproducibility is important; unfortunately, there is no mention of code release statement - even after acceptance.
- The code was omitted in the submission and thus evaluation of the soundness of the implementation could not be performed.

**Questions:**

- What do the authors mean by "test time" and how does that happen in practice? Do they run a benchmark? Keep track of computation time? I feel this requires some clarifications...
- Why the code was not included part of the assessment as a private artifact? I see no sensitive data and/or methods in the paper to warrant this.
- How do the clients balance the task? Is that something that is covered by NeFL? Does that balancing act happen once? Or is that performed dynamically over time?
- What is the impact of stragglers in the final output? What happens if a client drops from the computation?

---

> ### Author Response · Authors · 2023-11-14
> **Response to Reviewer HZdf**
>
> Dear Reviewer HZdf,
>
> We sincerely thank you for your positive and valuable feedback on our paper. We address your comments and questions below. In the revised draft, we take the reviewer's point and update the paper.  We mark our revision in $\color{blue}\text{blue}$.
>
> ---
> **1**. Clarification on test-time
>
> **Test-time denotes employing a model for inference after training.** After $N_s$ submodels have been trained, a client can run one of the submodels considering its current battery, runtime memory and available computing-power.
> - We added the explanation on test-time as a footnote in Section 4.
>
> > Wang, Dequan and Shelhamer, Evan and Liu, Shaoteng and Olshausen, Bruno and Darrell, Trevor. TENT: Fully Test-Time Adaptation by Entropy Minimization. In ICLR, 2021.
> ---
> **2**. Source code
>
> **We will include the github URL of our source code on the paper after blind review process.** We agree with the importance of reproducibility. Before the release, the reviewer can kindly refer to pseudo-code for parameter averaging provided in Appendix B.8.
> - We temporarily added ‘code will be available after blind review process’  in the abstract.
> ---
> **3**. Dynamical balancing
>
> **The clients are aware of their own dynamic status such as communication environment and available computing-power (run-time dynamics), respectively.** Then, the clients respectively request a NeFL server to transmit submodels for training that meet their current requirements in the communication round.
> - We revised the balancing procedure with more details in Section 4.
> ---
> **4**. Stragglers
>
> **Less stragglers output fast and stable convergence of FL with better classification accuracies while more stragglers output slow and worse convergence  with worse classification accuracies. Dropped clients wait until the next communication round.**
> The server can wait at every communication round for the slow client to complete its local updates and send parameters to a server. Alternatively, a server can wait until the pre-defined deadline (Reisizadeh et al., 2019) or can wait for a pre-defined number of clients every communication round (Nguyen et al., 2022).
> A fraction $C$ of clients is selected at the beginning of each round and the server sends the current global model to each of these clients (McMahan et al., 2017). Each client then performs local computation based on the submodel and its local dataset, and sends an update to the server.
> Referring to Algorithm 1, the clients in $\mathcal{C}_t$ are selected in the communication round $t$. We considered the clients’ dynamic environment detailed in Appendix B.7. In NeFL, the clients can be stragglers when they cannot even afford the smallest submodel.
> - We added the explanations on stragglers in Section 1.
> - We conducted additional experiments  across the different fraction rate with NeFL-WD (submodels that are scaled by both widthwise and depthwise) on ResNet18 from scratch. The larger fraction rate denotes the larger number of clients participate in the FL pipeline every round. We provide results that is averaged over three runs:
> | Fraction rate | Worst |  Avg  |
> |:----:|:-----:|:-----:|
> | 0.05 | 86.65 | 87.41 |
> | 0.1  | 86.82 | 87.84 |
> | 0.2  | 87.01 | 87.99 |
>
> > Amirhossein Reisizadeh, Hossein Taheri, Aryan Mokhtari, Hamed Hassani, Ramtin Pedarsani. Robust and Communication-Efficient Collaborative Learning. In NeurIPS, 2019.
>
> > John Nguyen, Kshitiz Malik, Hongyuan Zhan, Ashkan Yousefpour, Michael Rabbat, Mani Malek, Dzmitry Huba. Federated learning with buffered asynchronous aggregation. In AISTATS, 2022.
>
> > Brendan McMahan, Eider Moore, Daniel Ramage, Seth Hampson, and Blaise Aguera y Arcas. Communication-efficient learning of deep networks from decentralized data. In AISTATS, 2017.
> ---
> Thank you again for your valuable time and effort spent reviewing!

---

> > ### Comment · Reviewer_HZdf · 2023-11-23
> > **Read your rebuttal**
> >
> > Appreciate the response and the explanations. However, given tacking in account the other reviews, comments, and your rebuttal - I am keeping my score.

---

### Official Review · Reviewer_5ayx · 2023-11-02

**Soundness:** 4 excellent
**Presentation:** 3 good
**Contribution:** 4 excellent
**Rating:** 6
**Confidence:** 3

**Summary:**

The authors present nested federated learning (NeFL), a framework that divides the global model into multiple submodels using an interpretation of ODEs for depthwise scaling and continuous channel-based pruning on convolutional neural networks and node removal on fully connected neural networks for widthwise scaling, and performs aggregation using Nested Federated Averaging scheme for models with incompatible sizes.

**Strengths:**

++ The paper has carefully examined the related literature, and have observed the presence of low degrees of freedom in model architectures that were proposed in previous studies to combat heterogeneity by model splitting.

++ The paper has cleverly used findings from other papers to increase the credibility of their arguments.

++ The paper has performed a comprehensive set of experiments to show that NeFL performs significantly better than previous works.

**Weaknesses:**

-- Why scale the depth of the models using the interpretation of ODEs? The authors should motivate the reasons for using ODEs to scale depth and what makes this approach of scaling depth better than previous appraches. Why not use some other method for scaling depth? They should just perform depth scaling using ODEs and compare its performance to other depth scaling techniques in FL.

-- Since the authors propose a hybrid approach that combines depth scaling as well as width scaling in the spirit that a balanced network performs better, they should show how much better depth scaling using ODE is, when coupled with width scaling, and vice versa. Since in FL, the networks have not been depth scaled using ODEs, the authors should explain why they have not only used depth scaling via ODEs.

**Questions:**

-- How is the aggregation of the consistent parameters even meaningful? In the presence of non-IID datasets and ResNets of different depths and widths, each layer in a particular ResNet serves a different purpose than the corresponding layer in the other, ResNet i.e., layers L1, L2, and L3 with widths W1, W2 and W3 inside client A's Resenet with layers: (L1, L2, L3) will have different purposes than the corresponding layers L'1, L'2 and L'3 with widths W'1, W'2 and W'3 inside client B's Resenet with layers: (L'1, L'2, L'3, L'4, L'5). So what is the justification behind aggregating layers that are incoherent in terms of their purposes in the network?

-- The parameterAverage subroutine is expected to returns N_s submodels, however it does not compute submodels 2,...,N_{s-1}. The parameterAverage subroutine should compute and return the submodels 2,...,N_{s-1}: theta_{c,2},...,theta_{c,N_{s-1}}. For example theta_{c,N_{s-1}} = U over all {j<=N_{s-1}} phi_j.

-- The authors should explain their diagrams and algorithms thoroughly. Please refer to Writing Issues.


Writing Issues:

* The authors should explain Figure 2, part b as that would help the readers understand their aggregation scheme well.

* Algorithm 2 is difficult to read because the paper has used notations that were not pre-defined. For example in line 9, the superscript 'i' has not been pre-defined and it is hard to make sense of the backslash ('\').

---

> ### Author Response · Authors · 2023-11-14
> **Response to Reviewer 5ayx (1/2)**
>
> Dear Reviewer 5ayx,
>
> We sincerely thank you for your helpful and valuable feedback on our paper. We address your comments and questions below. In the revised draft, we take the reviewer's point and update the paper.  We mark our revision in $\color{blue}\text{blue}$.
>
> ---
> **1**. Interpretation of ODE for scaling
>
> We agree with the reviewer’s comments that there are other ways to scale down a model into submodels. **Our work is a general work that incorporates the previous depth-scaling methods.** For example, DepthFL denotes depthwise scaling without step size parameters and also has no decoupled inconsistent parameters.
>
> We provided the ablation study on Table 6 in Appendix A.3. NeFL-D denotes FL with scaling depthwise by our proposed method, NeFL-D (N/L) denotes FL with scaling depthwise without no learnable step parameters, NeFL-D_{O} denotes depthwise scaling with different initial values and NeFL-D_{O} (N/L) denotes depthwise scaling with different initial values without learnable step size parameters.
>
> ---
> **2**. Depth scaling coupled with width scaling
>
> **We provide ablation study of NeFL consisting of both depthwise and widthwise scaled submodels in Table 6 in Appendix A.3.** Furthermore, we provide the number of parameters and FLOPs for each method in Table 8. The results show that NeFL consisting of both depthwise and widthwise scaled submodels outperforms in ResNet18 with less FLOPs.
>
> **Meanwhile, our proposed depthwise scaling via ODE has been motivated by previous work that introduced learnable parameters multiplied to the output of residual blocks** (Touvron et al., 2021; Bachlechner et al., 2021; De & Smith, 2020). We interpreted the learnable parameters as the step sizes of an ODE solver. Our work includes the previous depth scaling methods.
>
> > Hugo Touvron, Matthieu Cord, Alexandre Sablayrolles, Gabriel Synnaeve, Hervé Jégou. Going deeper with Image Transformers. In ICCV 2021.
>
> > Thomas C. Bachlechner, Bodhisattwa Prasad Majumder, H. H. Mao, G. Cottrell, and Julian McAuley. Rezero is all you need: Fast convergence at large depth. In Uncertainty in AI (UAI), 2021.
>
> > Soham De and Samuel L Smith. Batch normalization biases residual blocks towards the identity function in deep networks. In NeurIPS, 2020.
>
> ---
> **3**. Aggregation of the consistent parameters
>
> We agree with the reviewer’s valuable comments. **Aggregating consistent parameters work as global representations while inconsistent parameters work as local refining representations.**
> Averaging consistent parameters of same model architecture trained by clients with non-IID local dataset can be improved by decoupling a few layers (Liang et al., 2020). In terms of aggregating different model architectures, the reviewer can refer to a study that each block in residual networks except the first block only slightly refine the features (Chang et al., 2018).
> Figure 3b on our manuscript ​​represents the L1 norm of weights averaged by the number of weights at each layer of five trained submodels. The submodel 1 which is the worst model, has a similar tendency of L1 norm to the widest model and the gap between submodel gets smaller as the model size gets wider. The slimmest model might have learned the most useful representation while additional parameters for larger models obtain still useful, but less useful information.
>
> > Bo Chang, Lili Meng, Eldad Haber, Frederick Tung, David Begert. Multi-level residual networks from dynamical systems view. In ICLR, 2018.
>
> > Paul Pu Liang, Terrance Liu, Liu Ziyin, Ruslan Salakhutdinov, and Louis-Philippe Morency. Think locally, act globally: Federated learning with local and global representations. arXiv preprint arXiv:2001.01523, 2020.
>
> ---
> **4**. N_s parameters
>
> $\theta_{c,k} \subset \theta_{c,N_s} \forall k$ that **a client can extract any parameters of a submodel from the largest submodel parameters**. Referring to Algorithm 1,  each client receives consistent parameters of the largest submodel and inconsistent parameters. The clients can dynamically determine which submodel to train for this round depending on the communication, computing, memory dynamics.
>
> - We added the statements ""Note that $\theta_{c,k} \subset \theta_{c,N_s}\forall k$." in “Section 4.2 Inconsistency”.
>
> ---
> **5**. Writing
>
> Thank you for your careful and considerate review. We take the reviewer's point and update the paper as follows:
> - We clarified and proofread descriptions and statements throughout our manuscript.
> - We provide notation for Algorithm 2 in “Section 4.2 Parameter averaging”.
> - We added details on the diagrams and algorithms.
>
> ---
> Thank you again for your valuable time and effort spent reviewing!

---

> ### Author Response · Authors · 2023-11-14
> **Response to Reviewer 5ayx (2/2)**
>
> **Ablations study**
> |                 | Depthwise scaling | Widthwise scaling | Adaptive step size |
> |:---------------:|:-----------------:|:-----------------:|:--------------------:|
> | DepthFL         | $\checkmark$        |                   |                    |
> | FjORD, HeteroFL |                   | $\checkmark$        |                    |
> | NeFL-D          | $\checkmark$        |                   | $\checkmark$         |
> | NeFL-W          |                   | $\checkmark$        | $\checkmark$         |
> | NeFL-WD         | $\checkmark$        | $\checkmark$        | $\checkmark$         |
> The results can be summarized by following statements:
> - NeFL-D shows better performance than NeFL-D_{O} that has larger initial values (Table 12) according to the number of skipped blocks. The rationale comes from the empirical results that trained step sizes are not as large as initial value for NeFL-D_{O} that large initial values for NeFL-D_{O} degrades the trainability of depthwise-scaled submodels.
> - Learnable step size parameters are effective for deep and pre-trained networks.
> - NeFL-D outperforms DepthFL that scales depthwise without inconsistent parameters.
> - NeFL-WD has less FLOPs compared to depthwise scaling and more FLOPs compared to widthwise scaling (Table 8).
> - NeFL-WD shows the better performance than NeFL-D on the worst submodel with ResNet18 and pre-trained ResNet18.
>
> We present the part of results in Appendix A.3. Below table is evaluated with pre-trained ResNet18:
>
> |        **Method**       | **Worst** | **Avg** |
> |:-----------------------:|:---------:|:-------:|
> | HeteroFL                | 78.26     | 84.06   |
> | FjORD                   | 86.37     | 88.91   |
> | **NeFL-W**              | 86.1      | 89.13   |
> | DepthFL                 | 47.76     | 82.85   |
> | NeFL-D (N/L)            | 86.95     | 89.77   |
> | NeFL-D$_\text{O}$ (N/L) | 86.24     | 89.76   |
> | **NeFL-D**              | 87.13     | 90.00   |
> | NeFL-D$_\text{O}$       | 87.02     | 89.72   |
> | **NeFL-WD**             | 88.61     | 89.60   |
> | NeFL-WD (N/L)           | 88.52     | 89.70   |
>
> |         | Widthwise/Depthwise scaling | Widthwise scaling | Depthwise scaling |
> |:-------:|:---------------------------:|:-----------------:|:-------------------:|
> | Param # | 6.71M                       | 6.71M             | 6.68M             |
> | FLOPs   | 87.8M                       | 85M               | 102M              |

---

### Author Response · Authors · 2023-11-14
**General Response**

Dear reviewers and AC,

We sincerely appreciate your valuable time and effort spent reviewing our manuscript.

As reviewers highlighted, our paper is well-motivated (ALL Reviewers) and we propose a simple yet novel and interesting method (Reviewer HZdf). Our paper has encouraging and comprehensive empirical results comparing a variety of recent frameworks (ALL Reviewers).

We appreciate your constructive comments on our manuscript. In response to the comments, we have carefully revised and enhanced the manuscript with the following additional discussions and experiments:

- Clarify contributions and motivations
- Clarify the details in Appendix A.3 presenting ablations study.
- Clarify descriptions and statements throughout our manuscript.

These updates are temporarily highlighted in “$\color{blue}\text{blue}$” for your convenience to check.

We hope our response and revision sincerely address all the reviewers’ concerns.

Thank you very much.

Best regards,

Authors.

---

### Meta-Review · Area_Chair_FGEg · 2023-12-09

**Metareview:**

This paper introduces an approach to enable federated training on devices with diverse capabilities, by using models with different width and height. Depth-wise and width-wise scaling are implemented by viewing the forward pass through a feed-forward network with skip connections as solving an ODE. The approach is validated via experiments on CIFAR-10.

The proposed approach addresses an important problem and initial results are promising.

The paper could be strengthened by more clearly illustrating in the paper how this work differs from previous work and by illustrating the benefits of the approach on a broader range of problem settings.

**Justification For Why Not Higher Score:**

Experiments only focus on a single, small task. Although multiple baselines are compared with the proposed approach, to be more convincing this really needs to be extended to workloads beyond CIFAR. The paper could also more convincingly explain the relationship and differences from prior work that lead to superior performance.

**Justification For Why Not Lower Score:**

N/A

---

### Decision · Program_Chairs · 2024-01-16

Reject